# A New Alternating Direction Method for Linear Programming

**Sinong Wang**
Department of ECE
The Ohio State University
wang.7691@osu.edu

**Ness Shroff**
Department of ECE and CSE
The Ohio State University
shroff.11@osu.edu

## Abstract

It is well known that, for a linear program (LP) with constraint matrix $\mathbf{A} \in \mathbb{R}^{m \times n}$, the Alternating Direction Method of Multiplier converges globally and linearly at a rate $O((\|\mathbf{A}\|_F^2 + mn)\log(1/\epsilon))$. However, such a rate is related to the problem dimension and the algorithm exhibits a slow and fluctuating "tail convergence" in practice. In this paper, we propose a new variable splitting method of LP and prove that our method has a convergence rate of $O(\|\mathbf{A}\|^2 \log(1/\epsilon))$. The proof is based on simultaneously estimating the distance from a pair of primal dual iterates to the optimal primal and dual solution set by certain residuals. In practice, we result in a new first-order LP solver that can exploit both the sparsity and the specific structure of matrix $\mathbf{A}$ and a significant speedup for important problems such as basis pursuit, inverse covariance matrix estimation, L1 SVM and nonnegative matrix factorization problem compared with the current fastest LP solvers.

## 1   Introduction

We are interested in applying the Alternating Direction Method of Multiplier (ADMM) to solve a linear program (LP) of the form

$$\min_{\mathbf{x} \in \mathbb{R}^n} \quad \mathbf{c}^T\mathbf{x} \quad s.t. \quad \mathbf{A}\mathbf{x} = \mathbf{b}, x_i \geq 0, i \in [n_b]. \tag{1}$$

where $\mathbf{c} \in \mathbb{R}^n$, $\mathbf{A} \in \mathbb{R}^{m \times n}$ is the constraint matrix, $\mathbf{b} \in \mathbb{R}^m$ and $[n_b] = \{1, \ldots, n_b\}$. This problem plays a major role in numerical optimization, and has been used in a large variety of application areas. For example, several important machine learning problems including the nonnegative matrix factorization (NMF) [1], $l_1$-regularized SVM [2], sparse inverse covariance matrix estimation (SICE) [3] and the basis pursuit (BP) [4], and the MAP inference [5] problem can be cast into an LP setting.

The complexity of the traditional LP solver is still at least quadratic in the problem dimension, i.e., the Interior Point method (IPM) with a weighted path finding strategy. However, many recent problems in machine learning have extremely large-scale targeting data but exhibit a sparse structure, i.e., $nnz(\mathbf{A}) \ll mn$, where $nnz(\mathbf{A})$ is the number of non-zero elements in the constraint matrix $\mathbf{A}$. This characteristic severely limits the ability of the IPM or Simplex technique to solve these problems. On the other hand, first-order methods have received extensive attention recently due to their ability to deal with large data sets. These methods require a matrix vector multiplication $\mathbf{A}\mathbf{x}$ in each iteration with complexity linear in $nnz(\mathbf{A})$. However, the key challenge in designing a first-order algorithm is that LPs are usually non-smooth and non-strongly convex optimization problems (may not have a unique solution). Utilizing the standard primal and dual stochastic sub-gradient descent method will result in an extremely slow convergence rate, i.e., $O(1/\epsilon^2)$ [6].

The ADMM was first developed in 1975 [7], and since then there have been several LP solvers based on this technique. Compared with the traditional Augmented Lagrangian Method (ALM), this

method splits the variable into several blocks, and optimizes the augmented Lagrangian (AL) function in a Gauss-Seidel fashion, which often results in relatively easier subproblems to solve. However, this method suffers from a slow convergence when the number of blocks increases. Moreover, the challenge of applying the ADMM to the LP is that the LP problem does not exhibit an explicit separable structure among variables, which are difficult to split in the traditional sense. The notable work [8] first applies the ADMM to solve the LP by augmenting the original $n$-dimensional variables into $nm-$dimensions, and the resultant Augmented Lagrangian function is separable among $n$ blocks of variables. They prove that this method converges globally and linearly. However, the rate of this method is dependent on the problem dimension $m, n$, and converges quite slowly when $m, n$ are large. Thus, they leave an open question on whether other efficient splitting methods exist, resulting in convergence analysis in the space with lower dimension $m$ or $n$.

In this paper, we propose a new splitting method for LP, which splits the equality and inequality constraints into two blocks. The resultant subproblems in each iteration are a linear system with a positive definite matrix, and $n$ one-dimensional truncation operations. We prove our new method converges globally and linearly at a faster rate compared with the method in [8]. Specifically, the main contributions of this paper can be summarized as follows: (i) We show that the existing ADMM in [8] exhibits a slow and fluctuating "tail convergence", and provide a theoretical understanding of why this phenomenon occurs. (ii) We propose a new ADMM method for LP and provide a new analysis of the linear convergence rate of this new method, which only involves $O(m + n)-$dimensional iterates. This result answers the open question proposed in [8]. (iii) We show that when the matrix $\mathbf{A}$ possesses some specific structure, the resultant subproblem can be solved in closed form. For the general constraint matrix $\mathbf{A}$, we design an efficiently implemented Accelerated Coordinate Descent Method (ACDM) to solve the subproblem in $O(\log(1/\epsilon)nnz(\mathbf{A}))$ time. (iv) Practically, we show that our proposed algorithm significantly speeds up solving the basis pursuit, $l_1$-regularized SVM, sparse inverse covariance matrix estimation, and the nonnegative matrix factorization problem compared with existing splitting method [8] and the current fastest first-order LP solver in [9].

## 2 Preliminaries

In this section, we first review several definitions that will be used in the sequel. Then we illustrate some observations from the existing method. We also include several LP-based machine problems that can be cast into the LP setting in the Appendix.

### 2.1 Notation

A twice differentiable function $f : \mathbb{R}^n \to \mathbb{R}$ has strong convexity parameter $\sigma$ if and only if its Hessian satisfies $\nabla^2 f(\mathbf{x}) \succeq \sigma \mathbf{I}, \forall \mathbf{x}$. We use $\|\cdot\|$ to denote standard $l_2$ norm for vector or spectral norm for matrix, $\|\cdot\|_1$ to denote the $l_1$ norm and $\|\cdot\|_F$ to denote the Frobenius norm. A twice differentiable function $f : \mathbb{R}^n \to \mathbb{R}$ has a component-wise Lipschitz continuous gradient with constant $L_i$ if and only if $\|\nabla_i f(\mathbf{x}) - \nabla_i f(\mathbf{y})\| \le L_i \|\mathbf{x} - \mathbf{y}\|, \forall \mathbf{x}, \mathbf{y}$. For example, for the quadratic function $F(\mathbf{x}) = \frac{1}{2}\|\mathbf{A}\mathbf{x} - \mathbf{b}\|^2$, the gradient $\nabla F(\mathbf{x}) = \mathbf{A}^T(\mathbf{A}\mathbf{x} - \mathbf{b})$ and the Hessian $\nabla^2 F(\mathbf{x}) = \mathbf{A}^T \mathbf{A}$. Hence the parameter $\sigma$ and $L_i$ satisfy (choose $\mathbf{y} = \mathbf{x} + t\mathbf{e}_i$, where $t \in \mathbb{R}$, $\mathbf{e}_i \in \mathbb{R}^n$ is the unit vector), $\mathbf{x}\mathbf{A}^T \mathbf{A}\mathbf{x} \ge \sigma \|\mathbf{x}\|^2$ and $t\mathbf{A}_i^T \mathbf{A}\mathbf{e}_i \le L_i |t|, \forall \mathbf{x}, t$. Thus, the $\sigma$ is the smallest eigenvalue of $\mathbf{A}^T \mathbf{A}$ and $L_i = \|\mathbf{A}_i\|^2$, where $\mathbf{A}_i$ is the $i$th column of the matrix $\mathbf{A}$. The projection operator of point $\mathbf{x}$ into convex set $\mathcal{S}$ is defined as $[\mathbf{x}]_{\mathcal{S}} = \arg\min_{\mathbf{u} \in \mathcal{S}} \|\mathbf{x} - \mathbf{u}\|$. If $\mathcal{S}$ is the non-negative cone, let $[\mathbf{x}]_+ \triangleq [\mathbf{x}]_{\mathcal{S}}$. Let $V_i = [0, \infty)$ for $i \in [n_b]$ and $V_i = \mathbb{R}$ for $i \in [n_f]$.

### 2.2 Tail Convergence of the Existing ADMM Method

The existing ADMM in [8] solves the LP (1) by following procedure: in each iteration $k$, go through the following two steps:

1. Primal update: $x_i^{k+1} = \left[ x_i^k + \frac{1}{\|\mathbf{A}_i\|^2} \left( \frac{\mathbf{A}_i^T(\mathbf{b} - \mathbf{A}\mathbf{x}^k)}{q} - \frac{c_i - \mathbf{A}_i^T \mathbf{z}^k}{\lambda} \right) \right]_{V_i}, i = 1, \ldots, n.$

2. Dual update: $\mathbf{z}^{k+1} = \mathbf{z}^k - \frac{\lambda}{q}(\mathbf{A}\mathbf{x}^k - \mathbf{b}).$

We plot the solving accuracy versus the number of iterations for solving three kinds of problems (see Fig.1 in Appendix). We can observe that it converges fast in the initial phase, but exhibits a slow and

fluctuating convergence when the iterates approach the optimal set. This method originates from a specific splitting method in the standard $2-$block ADMM [10]. To provide some understanding of this phenomenon, we show that this method can be actually recovered by an inexact Uzawa method [11]. The Augmented Lagrangian function of the problem (1) is denoted by $L(\mathbf{x}, \mathbf{z}) = \mathbf{c}^T \mathbf{x} + \frac{\rho}{2} \|\mathbf{Ax} - \mathbf{b} - \mathbf{z}/\rho\|^2$. In each iteration $k$, the inexact Uzawa method first minimizes a local second-order approximation of the quadratic term in $L(\mathbf{x}, \mathbf{z}^k)$ with respect to primal variables $\mathbf{x}$, specifically,

$$\mathbf{x}^{k+1} = \arg \min_{x_i \in V_i} \mathbf{c}^T \mathbf{x} + \langle \rho \mathbf{A}^T (\mathbf{Ax}^k - \mathbf{b} - \mathbf{z}^k/\rho), \mathbf{x} - \mathbf{x}^k \rangle + \frac{1}{2} \|\mathbf{x} - \mathbf{x}^k\|_{\mathbf{D}}, \qquad (2)$$

then update the dual variables by $\mathbf{z}^{k+1} = \mathbf{z}^k - \rho(\mathbf{Ax}^{k+1} - \mathbf{b})$. Let the proximity parameter $\rho = \lambda/q$ and matrix $\mathbf{D}$ equal to the diagonal matrix $\text{diag}\{\ldots, 1/q\|\mathbf{A}_i\|^2, \ldots\}$, then we can recover the above algorithm by the first-order optimality condition of (2). This equivalence allows us to illustrate the main reason for the slow and fluctuating "tail convergence" comes from the *inefficiency of such a local approximation of the Augmented Lagrangian function when the iterates approach the optimal set*.

One straightforward idea to resolve this issue is to minimize the Augmented Lagrangian function exactly instead of its local approximation, which leads to the classic ALM. There exists a line of works focusing on analyzing the convergence of applying ALM to LP [9, 12, 13]. This method will produce a sequence of constrained quadratic programs (QP) that are difficult to solve. The work [9] proves that the proximal Coordinate Descent method can solve each QPs at a linear rate even when matrix $\mathbf{A}$ is not full column rank. However, there exists several drawbacks in this approach: (i) the practical solving time of each subproblem is quite long when $\mathbf{A}$ is rank-deficient; (ii) the theoretical performance and complexity of using recent accelerated techniques in proximal optimization [14] with the ALM is unknown; (iii) it cannot exploit the specific structure of matrix $\mathbf{A}$ when solving each constrained QP. Therefore, it motivates us to investigate the new and efficient variable splitting method for such a problem.

## 3 New Splitting Method in ADMM

We first separate the equality and inequality constraints of the above LP (1) by adding another group of variables $\mathbf{y} \in \mathbb{R}^n$.

$$\min \quad \mathbf{c}^T \mathbf{x} \qquad (3)$$
$$s.t. \quad \mathbf{Ax} = \mathbf{b}, \mathbf{x} = \mathbf{y},$$
$$\quad y_i \geq 0, i \in [n_b].$$

The dual of problem (3) takes the following form.

$$\min \quad \mathbf{b}^T \mathbf{z}_x \qquad (4)$$
$$s.t. \quad -\mathbf{A}^T \mathbf{z}_x - \mathbf{z}_y = \mathbf{c},$$
$$\quad z_{y,i} \leq 0, i \in [n_b], z_{y,i} = 0, i \in [n] \backslash [n_b].$$

Let $\mathbf{z}_x, \mathbf{z}_y$ be the Lagrange multipliers for constraints $\mathbf{Ax} = \mathbf{b}$, $\mathbf{x} = \mathbf{y}$, respectively. Define the indicator function $g(\mathbf{y})$ of the non-negative cone: $g(\mathbf{y}) = 0$ if $y_i \geq 0, \forall i \in [n_b]$; otherwise $g(\mathbf{y}) = +\infty$. Then the augmented Lagrangian function of the primal problem (3) is defined as

$$L(\mathbf{x}, \mathbf{y}, \mathbf{z}) = \mathbf{c}^T \mathbf{x} + g(\mathbf{y}) + \mathbf{z}^T (\mathbf{A}_1 \mathbf{x} + \mathbf{A}_2 \mathbf{y} - \overline{\mathbf{b}}) + \frac{\rho}{2} \|\mathbf{A}_1 \mathbf{x} + \mathbf{A}_2 \mathbf{y} - \overline{\mathbf{b}}\|^2, \qquad (5)$$

where $\mathbf{z} = [\mathbf{z}_x; \mathbf{z}_y]$. The matrix $\mathbf{A}_1$, $\mathbf{A}_2$ and vector $\overline{\mathbf{b}}$ are denoted by

$$\mathbf{A}_1 = \begin{bmatrix} \mathbf{A} \\ \mathbf{I} \end{bmatrix}, \mathbf{A}_2 = \begin{bmatrix} \mathbf{0} \\ -\mathbf{I} \end{bmatrix}, \text{ and } \overline{\mathbf{b}} = \begin{bmatrix} \mathbf{b} \\ \mathbf{0} \end{bmatrix}. \qquad (6)$$

In each iteration $k$, the standard ADMM go through following three steps:

1. Primal update: $\mathbf{x}^{k+1} = \arg \min_{\mathbf{x} \in \mathbb{R}^n} L(\mathbf{x}, \mathbf{y}^k, \mathbf{z}^k)$.

2. Primal update: $\mathbf{y}^{k+1} = \arg \min_{\mathbf{y} \in \mathbb{R}^n} L(\mathbf{x}^{k+1}, \mathbf{y}, \mathbf{z}^k)$.

**Algorithm 1** Alternating Direction Method of Multiplier with Inexact Subproblem Solver

---
Initialize $\mathbf{z}^0 \in \mathbb{R}^{m+n}$, choose parameter $\rho > 0$.
**repeat**
    1. Primal update: find $\mathbf{x}^{k+1}$ such that $F_k(\mathbf{x}^{k+1}) - \min_{\mathbf{x} \in \mathbb{R}^n} F_k(\mathbf{x}) \leq \epsilon_k$.
    2. Primal update: for each $i$, let $y_i^{k+1} = \left[ x_i^{k+1} + z_{y,i}^k/\rho \right]_{V_i}$.
    3. Dual update: $\mathbf{z}_x^{k+1} = \mathbf{z}_x^k + \rho(\mathbf{A}\mathbf{x}^{k+1} - \mathbf{b})$, $\mathbf{z}_y^{k+1} = \mathbf{z}_y^k + \rho(\mathbf{x}^{k+1} - \mathbf{y}^{k+1})$.
**until** $\|\mathbf{A}\mathbf{x}^{k+1} - \mathbf{b}\|_\infty \leq \epsilon$ and $\|\mathbf{x}^{k+1} - \mathbf{y}^{k+1}\|_\infty \leq \epsilon$

---

    3. Dual update: $\mathbf{z}^{k+1} = \mathbf{z}^k + \rho(\mathbf{A}_1 \mathbf{x}^{k+1} + \mathbf{A}_2 \mathbf{y}^{k+1} - \overline{\mathbf{b}})$.

The first step is an unconstrained quadratic program, which can be simplified as

$$\mathbf{x}^{k+1} = \arg\min_{\mathbf{x}} F_k(\mathbf{x}) \triangleq \mathbf{c}^T \mathbf{x} + (\mathbf{z}^k)^T \mathbf{A}_1 \mathbf{x} + \frac{\rho}{2} \|\mathbf{A}_1 \mathbf{x} + \mathbf{A}_2 \mathbf{y}^k - \overline{\mathbf{b}}\|^2. \tag{7}$$

The gradient of the function $F_k(\mathbf{x})$ can be expressed as

$$\nabla F_k(\mathbf{x}) = \rho(\mathbf{A}^T \mathbf{A} + \mathbf{I})\mathbf{x} + \mathbf{A}_1^T[\mathbf{z}^k + \rho(\mathbf{A}_2 \mathbf{y}^k - \overline{\mathbf{b}})] + \mathbf{c}, \tag{8}$$

and the Hessian of function $F_k(\mathbf{x})$ is

$$\nabla^2 F_k(\mathbf{x}) = \rho(\mathbf{A}^T \mathbf{A} + \mathbf{I}). \tag{9}$$

Further, based on the first-order optimality condition, the first step is equivalent to solving a linear system, which requires inverting the Hessian matrix (9). In practice, the complexity is quite high to be exactly solved unless the Hessian exhibits some specific structures. Thus, we relax the first step into the inexact minimization: find $\mathbf{x}^{k+1}$ such that

$$F_k(\mathbf{x}^{k+1}) - \min_{\mathbf{x} \in \mathbb{R}^n} F_k(\mathbf{x}) \leq \epsilon_k, \tag{10}$$

where $\epsilon_k$ is the given accuracy. Transforming the indicator function $g(\mathbf{y})$ back to the constraints, the second step can be separated into $n$ one−dimensional optimization problems: for each $i$,

$$y_i^{k+1} = \arg\min_{y_i \in V_i} -z_{y,i}^k y_i + \frac{\rho}{2}(y_i - x_i^{k+1})^2 = \left[ x_i^{k+1} + z_{y,i}^k/\rho \right]_{V_i}.$$

The resultant algorithm is sketched in Algorithm 1. In some applications such as $l_1$-regularized SVMs and basis pursuit problem, the objective function contains the $l_1$ norm of the variables. Transforming to the canonical form (1) will introduce additional $n$ variables and $2n$ constraints. One important feature in our method is that we can split the objective function by adding variable $\mathbf{y}$. The corresponding subproblems are similar with Algorithm 1 and the only difference is that the second step will be $n$ one−dimensional shrinkage operations. (Details can be seen in Appendix.)

## 4 Convergence Analysis of New ADMM

In this section, we prove that the Algorithm 1 converges at a global and linear rate, and provide a roadmap of the main technical development. We can first write the primal problem (3) as the following standard 2−block form.

$$\min_{\mathbf{x},\mathbf{y}} f(\mathbf{x}) + g(\mathbf{y}) \quad s.t. \quad \mathbf{A}_1 \mathbf{x} + \mathbf{A}_2 \mathbf{y} = \overline{\mathbf{b}}, \tag{11}$$

where $f(\mathbf{x}) = \mathbf{c}^T \mathbf{x}$ and $g(\mathbf{y})$ is the indicator function as defined before. Most works in the literature prove that the 2-block ADMM converges globally and linearly via assuming that one of the functions $f$ and $g$ is strongly convex [15, 16, 17]. Unfortunately, both the linear function $f$ and the indicator function $g$ in the LP do not satisfy this property, which poses a significant challenge on the current analytical framework. There exists several recent works trying to address this problem in some sense. In work [18], they have demonstrated that when the dual step size $\rho$ is sufficiently small (impractical), the ADMM converges globally linearly, while no implicit rate is given. The work [13] shows that the ADMM is locally linearly converged when applying to LP. They utilize a unique combination of iterates and conduct a spectral analysis. However, they still leave an open question whether ADMM converges globally and linearly when applying to the LP in the above form.

In the sequel, we will answer this question positively and provide an accurate analysis of such a splitting method. The main technical development is based on a geometric argument: we first prove that the set formed by optimal primal and dual solutions of LP (3) is a $(3n + m)-$dimensional polyhedron $\mathcal{S}^*$; then we utilize certain global error bound to simultaneously estimate the distance from iterates $\mathbf{x}^{k+1}, \mathbf{y}^k, \mathbf{z}^k$ to $\mathcal{S}^*$. All detailed proofs are given in the Appendix.

**Lemma 1.** (Convergence of 2-block ADMM [10]) *Let* $\mathbf{p}^k = \mathbf{z}^k - \rho\mathbf{A}_2\mathbf{y}^k$, *we have*

$$\|\mathbf{p}^{k+1} - [\mathbf{p}^{k+1}]_{G^*}\|^2 \le \|\mathbf{p}^k - [\mathbf{p}^k]_{G^*}\|^2 - \|\mathbf{p}^{k+1} - \mathbf{p}^k\|^2,$$

*where* $G^* \triangleq \{\mathbf{p}^* \in \mathbb{R}^{m+n}|T(\mathbf{p}^*) = \mathbf{p}^*\}$, *and the definition of operator* $T$ *is given in (54) in Appendix. Moreover, if the LP (3) has a pair of optimal primal and dual solution, the iterates* $\mathbf{x}^k, \mathbf{y}^k$ *and* $\mathbf{z}^k$ *converges to an optimal solution; Otherwise, at least one of the iterates is unbounded.*

Lemma 1 is tailored from applying the classic Douglas-Rachford splitting method to the LP. This result guarantees that the sequence $\mathbf{p}^k$ produced by ADMM globally converges under a mild assumption. However, to establish the linear convergence rate, the key lies in estimating the other side inequality,

$$\|\mathbf{p}^k - [\mathbf{p}^k]_{G^*}\| \le \gamma\|\mathbf{p}^{k+1} - \mathbf{p}^k\|, \gamma > 0. \tag{12}$$

Then one can combine these two results together to prove that sequence $\mathbf{p}^k$ converges globally and linearly with $\|\mathbf{p}^{k+1} - [\mathbf{p}^{k+1}]_{G^*}\|^2 \le (1 - 1/\gamma^2) \cdot \|\mathbf{p}^k - [\mathbf{p}^k]_{G^*}\|^2$, which further can be used to show the $R-$linear convergence of iterates $\mathbf{x}^k, \mathbf{y}^k$ and $\mathbf{z}^k$. To estimate the constant $\gamma$, we first describe the geometry formed by the optimal primal solutions $\mathbf{x}^*, \mathbf{y}^*$ and dual solutions $\mathbf{z}^*$ of the LP (3).

**Lemma 2.** (Geometry of the optimal solution set of LP) *The variables* $(\mathbf{x}^*, \mathbf{y}^*)$ *are the optimal primal solutions and* $\mathbf{z}^*$ *are optimal dual solutions of LP (3) if and only if (i)* $\mathbf{A}\mathbf{x}^* = \mathbf{b}$, $\mathbf{x}^* = \mathbf{y}^*$; *(ii)* $-\mathbf{A}^T\mathbf{z}_{\mathbf{x}}^* - \mathbf{z}_y^* = \mathbf{c}$; *(iii)* $y_i^* \ge 0, z_{y,i}^* \le 0, i \in [n_b]$; $z_{y,i}^* = 0, i \in [n]\backslash[n_b]$; *(iv)* $\mathbf{c}^T\mathbf{x}^* + \mathbf{b}^T\mathbf{z}_x^* = 0$.

In Lemma 2, one interesting element is to utilize the strong duality condition (iv) to eliminate the complementary slackness in the standard KKT condition. Then, the set of optimal primal and dual solutions is described only by affine constraints, which further implies that the optimal solution set is an $(m + 3n)-$dimensional polyhedron. We use $\mathcal{S}^*$ to denote such a polyhedron.

**Lemma 3.** (Hoffman bound [19, 20]) *Consider a polyhedron set* $\mathcal{S} = \{\mathbf{x} \in \mathbb{R}^d|\mathbf{E}\mathbf{x} = \mathbf{t}, \mathbf{C}\mathbf{x} \le \mathbf{d}\}$. *For any point* $\mathbf{x} \in \mathbb{R}^d$, *we have*

$$\|\mathbf{x} - [\mathbf{x}]_{\mathcal{S}}\| \le \theta_S \left\| \begin{bmatrix} \mathbf{E}\mathbf{x} - \mathbf{t} \\ [\mathbf{C}\mathbf{x} - \mathbf{d}]_+ \end{bmatrix} \right\|, \tag{13}$$

*where* $\theta_S$ *is the Hoffman constant that depends on the structure of polyhedron* $\mathcal{S}$.

According to the result in Lemma 2, it seems that we can use the Hoffman bound to estimate the distance between the current iterates $(\mathbf{x}^k, \mathbf{y}^k, \mathbf{z}^k)$ and the solution set $\mathcal{S}^*$ via the their primal and dual residual. However, to obtain the form of inequality (12), we need to bound such a residual in terms of $\|\mathbf{p}^k - \mathbf{p}^{k+1}\|$. Indeed, we have these results.

**Lemma 4.** (Estimation of residual) *The sequence* $(\mathbf{x}^{k+1}, \mathbf{y}^k, \mathbf{z}^k)$ *produced by Algorithm 1 satisfies*

$$\begin{cases} \mathbf{A}_1\mathbf{x}^{k+1} + \mathbf{A}_2\mathbf{y}^k - \overline{\mathbf{b}} = (\mathbf{p}^{k+1} - \mathbf{p}^k)/\rho, \\ \mathbf{c} + \mathbf{A}_1^T\mathbf{z}^k = \mathbf{A}_1^T(\mathbf{p}^k - \mathbf{p}^{k+1}), \\ \mathbf{c}^T\mathbf{x}^{k+1} + \mathbf{b}^T\mathbf{z}_x^k = (\mathbf{A}_1\mathbf{x}^{k+1} - \mathbf{z}^k/\rho)^T(\mathbf{p}^k - \mathbf{p}^{k+1}), \\ y_i^k \ge 0, z_{y,i}^k \le 0, i \in [n_b]; z_{y,i}^k = 0, i \in [n]\backslash[n_b]. \end{cases}$$

One observation from Lemma 4 is that Algorithm 1 automatically preserves the boundness and the complementary slackness of both primal and dual iterates. Instead, in the previous algorithm in [8], the complementary slackness is not preserved during the iteration. Combining the results in Lemma 2, Lemma 3 and Lemma 4, we are readily to estimate the constant $\gamma$.

**Lemma 5.** (Estimation of linear rate) *The sequence* $\mathbf{p}^k = \mathbf{z}^k - \rho\mathbf{A}_2\mathbf{y}_k$ *produced by Algorithm 1 satisfies* $\|\mathbf{p}^k - [\mathbf{p}^k]_{G^*}\| \le \gamma\|\mathbf{p}^{k+1} - \mathbf{p}^k\|$, *where the rate* $\gamma$ *is given by*

$$\gamma = (1 + \rho)\left[\frac{R_z + 1}{\rho} + R_x\|\mathbf{A}_1\| + \|\mathbf{A}_1^T\|\right]\theta_{S^*}. \tag{14}$$

$R_x = \sup_k \|\mathbf{x}^k\| < +\infty$, $R_z = \sup_k \|\mathbf{z}^k\| < +\infty$ *are the maximum radius of iterates* $\mathbf{x}^k$ *and* $\mathbf{z}^k$.

Then we can establish the global and linear convergence of Algorithm 1.

**Theorem 1.** (Linear convergence of Algorithm 1) *Denote $\mathbf{z}^k$ as the primal iterates produced by Algorithm 1. To guarantee that there exists an optimal dual solution $\mathbf{z}^*$ such that $\|\mathbf{z}^k - \mathbf{z}^*\| \leq \epsilon$, it suffices to run Algorithm 1 for number of iterations $K = 2\gamma^2 \log(2D_0/\epsilon)$ with the solving accuracy $\epsilon_k$ satisfying $\epsilon_k \leq \epsilon^2/8K^2$, where $D_0 = \|\mathbf{p}^0 - [\mathbf{p}^0]_{G^*}\|$.*

The proof of Theorem 1 consists of two steps: first, we establish the global and linear convergence rate of Algorithm 1 when $\epsilon_k = 0, \forall k$ (exact subproblem solver); then we relax this condition and prove that when $\epsilon_k$ is less than a specified threshold, the algorithm still shares a convergence rate of the same order. The results of primal iterates $\mathbf{x}^k$ and $\mathbf{y}^k$ are similar.

## 5   Efficient Subproblem Solver

In this section, we will show that, due to our specific splitting method, each subproblem in line 1 of Algorithm 1 can be either solved in closed-form expression or efficiently solved by the Accelerated Coordinate Descent Method.

### 5.1   Well-structured Constraint Matrix

Let the gradient (8) vanish, then the primal iterates $\mathbf{x}^{k+1}$ can be exactly determined by
$$\mathbf{x}^{k+1} = \rho^{-1}(\mathbf{I} + \mathbf{A}^T\mathbf{A})^{-1}\mathbf{d}^k, \text{ with } \mathbf{d}^k = -\mathbf{A}_1^T[\mathbf{z}^k + \rho(\mathbf{A}_2\mathbf{y}^k - \overline{\mathbf{b}})] - \mathbf{c}, \tag{15}$$
which requires inverting an $n \times n$ positive definite matrix $\mathbf{I} + \mathbf{A}^T\mathbf{A}$, or equivalently, inverting an $m \times m$ positive definite matrix $\mathbf{I} + \mathbf{A}\mathbf{A}^T$ via the following Sherman–Morrison–Woodbury identity,
$$(\mathbf{I} + \mathbf{A}^T\mathbf{A})^{-1} = \mathbf{I} - \mathbf{A}^T(\mathbf{I} + \mathbf{A}\mathbf{A}^T)^{-1}\mathbf{A}. \tag{16}$$
One basic fact is that we only need to invert such a matrix once and then use this cached factorization in subsequent iterations. Therefore, there are several cases for which the above factorization can be efficiently calculated: (i) Factorization has a closed-form expression. For example, in the LP-based MAP inference [5], the matrix $\mathbf{I} + \mathbf{A}^T\mathbf{A}$ is block diagonal, and each block has been shown to possess a closed-form factorization. Another important application is that, in the basis pursuit problem, the encoding matrices such as DFT (discrete Fourier transform) and DWHT (discrete Walsh-Hadamard transform) matrices have orthonormal rows and satisfy $\mathbf{A}\mathbf{A}^T = \mathbf{I}$. Based on (15), each $\mathbf{x}^{k+1} = \rho^{-1}(\mathbf{I} - \frac{1}{2}\mathbf{A}^T\mathbf{A})\mathbf{d}^k$ and can be calculated in $O(n\log(n))$ time by certain fast transforms. (ii) Factorization has a low-complexity: the dimension $m$ (or $n$) is small, i.e., $m = 10^4$. Such a factorization can be calculated in $O(m^3)$ and the complexity of each iteration is only $O(nnz(\mathbf{A}) + m^2)$. Detailed applications can be viewed in Appendix.

**Remark 1.** *In the traditional Augmented Lagrangian method, the resultant subproblem is a constrained and non-strongly convex QP (Hessian is not invertible), which does not allow the above close-form expression. Besides, in the ALCD [9], the coordinate descent (CD) step only picks one column in each iteration and cannot exploit the nice structure of matrix $\mathbf{A}$. One idea is to modify the CD step in [9] to the proximal gradient descent. However, it will greatly increase the computation time due to the large number of inner gradient descent steps.*

### 5.2   General Constraint Matrix

However, in other applications, the constraint matrix $\mathbf{A}$ only exhibits the sparsity, which is difficult to invert. To resolve this issue, we resort to the current fastest accelerated coordinate descent method [21]. This method has an order improvement up to $O(\sqrt{n})$ of iteration complexity compared with previous accelerated coordinate descent methods [22]. However, the naive evaluation of partial derivative of function $F_k(\mathbf{x})$ in ACDM takes $O(nnz(\mathbf{A}))$ time; second, the time cost of full vector operation in each iteration of ACDM is $O(n)$. We will show that these difficulties can be tackled by a carefully designed implementation technique[1] and the main procedure is listed in Algorithm 2. Here the iterates $\mathbf{s_t}$ and matrix $\mathbf{M}$ in Algorithm 2 is defined as

$$\mathbf{M} = \begin{bmatrix} 1 - \alpha_v & \alpha_v \\ \beta_u & 1 - \beta_u \end{bmatrix} \text{ with } \begin{bmatrix} \alpha_v \\ \beta_u \end{bmatrix} = \begin{bmatrix} \frac{\tau}{1+\eta\rho} \\ \frac{\eta\beta\rho}{1+\eta\rho} \end{bmatrix} \text{ and } \mathbf{s}_t^i = \begin{bmatrix} \left( \frac{\eta\tau}{p_i(1+\eta\rho)} + \frac{1-\tau}{L_i} \right) \nabla_i F_k(\mathbf{u}_t)\mathbf{e}_i^T \\ \frac{\eta}{p_i(1+\eta\rho)} \nabla_i F_k(\mathbf{u}_t)\mathbf{e}_i^T \end{bmatrix},$$
$$\tag{17}$$

**Algorithm 2** Efficiently Subproblem Solver

---

Initialize $\mathbf{u}_0, \mathbf{v}_0, \overline{\mathbf{u}}_0 = \mathbf{A}\mathbf{u}_0, \overline{\mathbf{v}}_0 = \mathbf{A}\mathbf{v}_0$, matrix $\mathbf{M}$, parameter $\tau, \eta, S$ by (17) and distribution $p = [\ldots, \sqrt{1 + \|\mathbf{A}_i\|^2}/S, \ldots]$ and let $\mathbf{d}^k = \mathbf{A}_1^T[\mathbf{z}^k + \rho(\mathbf{A}_2\mathbf{y}^k - \overline{\mathbf{b}})] + \mathbf{c}$.
**repeat**
   $[\mathbf{u}_t, \mathbf{v}_t]^T = \mathbf{M}_{t-1} \cdot [\overline{\mathbf{u}}, \overline{\mathbf{v}}]^T$ and $[\overline{\mathbf{u}}_t, \overline{\mathbf{v}}_t]^T = \mathbf{M}_{t-1} \cdot [\overline{\mathbf{u}}, \overline{\mathbf{v}}]^T$.
   Sample $i$ from $[n]$ based on probability distribution $p$.
   $\nabla_i F_k(\mathbf{u}_t) = \rho(\mathbf{A}_i)^T\overline{\mathbf{u}}_t + \rho u_{t,i} + d_i^k$, and calculate $\mathbf{s}_t^i$ by (17).
   $\mathbf{M}_t = \mathbf{M} \cdot \mathbf{M}_{t-1}$. Update $\begin{bmatrix}\mathbf{u}^T \\ \mathbf{v}^T\end{bmatrix} = \begin{bmatrix}\mathbf{u}^T \\ \mathbf{v}^T\end{bmatrix} - \mathbf{M}_t^{-1}\mathbf{s}_t^i$, $\begin{bmatrix}\overline{\mathbf{u}}^T \\ \overline{\mathbf{v}}^T\end{bmatrix} = \begin{bmatrix}\overline{\mathbf{u}}^T \\ \overline{\mathbf{v}}^T\end{bmatrix} - \mathbf{M}_t^{-1}\mathbf{s}_t^i\mathbf{A}^T$,
**until** Converge
Output $\mathbf{x}^{k+1} = (\mathbf{u}_T - \tau\mathbf{v}_T)/(1 - \tau)$.

---

where $\eta = \frac{1}{\tau S^2}, \tau = \frac{2}{1+\sqrt{4S^2/\rho+1}}, S = \sum_{i=1}^n \sqrt{\|\mathbf{A}_i\|^2 + 1}$. See more details in Appendix.

**Lemma 6.** (Inner complexity) *In each iteration of Algorithm 2, if the current picked coordinate is $i$, the update can be finished in $O(nnz(\mathbf{A}_i))$ time, moreover, to guarantee that $F_k(\mathbf{x}^{k+1}) - \min_\mathbf{x} F_k(\mathbf{x}) \leq \epsilon_k$ with probability $1 - p$, it suffices to run Algorithm 2 for number of iterations*

$$T_k \geq O(1) \cdot \sum_{i=1}^n \|\mathbf{A}_i\| \log\left(\frac{D_0^k}{\epsilon_k p}\right), \quad D_0^k = \|\overline{F}_k(\mathbf{u}_0) - \min_\mathbf{x} \overline{F}_k(\mathbf{x})\|. \tag{18}$$

The above iteration complexity is obtained by choosing parameter $\beta = 0$ in [21] and utilizing the Theorem 1 in [23] to transform the convergence in expectation to the form of probability.

**Theorem 2.** (Overall complexity) *Denote $\mathbf{z}^k$ as the dual iterates produced by Algorithm 1. To guarantee that there exists an optimal solution $\mathbf{z}^*$ such that $\|\mathbf{z}^k - \mathbf{z}^*\| \leq \epsilon$ with probability $1 - p$, it suffices to run Algorithm 1 for $k \geq 2\gamma^2 \log(2D_0/\epsilon)$ outer iterations and solve each sub-problem (7) for the number of inner iterations*

$$T \geq O(1) \cdot \sum_{i=1}^n \|\mathbf{A}_i\| \log\left(\frac{\rho(D_0^k)^{\frac{1}{3}}\gamma^2}{\epsilon^{\frac{2}{3}}p^{\frac{1}{3}}} \log\left(\frac{2D_0}{\epsilon}\right)\right). \tag{19}$$

The results for the primal iterates $\mathbf{x}^k$ and $\mathbf{y}^k$ are similar. In the existing ADMM [8], each primal and dual update only requires $O(nnz(\mathbf{A}))$ time to solve. The complexity of this method is

$$O(a_m\mu^2(a_mR_x + d_mR_z)^2(\sqrt{mn} + \|\mathbf{A}\|_F)^2 nnz(\mathbf{A})\log(1/\epsilon)),$$

where $a_m = \max_i \|\mathbf{A}_i\|$, $d_m$ is the largest number of non-zero elements of each row of matrix $\mathbf{A}$, and $\mu$ is the Hoffman constant depends on the optimal solution set of LP. Based on Theorem 2, an estimation of the worst-case complexity of Algorithm 1 is

$$O(a_m\theta_{S^*}^2(R_x\|\mathbf{A}\| + R_z)^2 nnz(\mathbf{A})\log^2(1/\epsilon)).$$

Remark that our method has a weak dependence on the problem dimension compared with the existing ADMM. Since the Frobenius norm of a matrix satisfies $\|\mathbf{A}\|_2 \leq \|\mathbf{A}\|_F$, our method is faster than the one in [8].

## 6 Numerical Results

In this section, we examine the performance of our algorithm and compare it with the state-of-art of algorithms developed for solving the LP. The first is the existing ADMM in [8]. The second is the ALCD method in [9], which is reported to be the current fastest first-order LP solver. They have shown that this algorithm can significantly speed up solving several important machine learning problems compared with the Simplex and IPM. We name our Algorithm 1 as LPADMM. In the experiments, we require that the accuracy of subproblem solver $\epsilon_k = 10^{-3}$ and the stopping criteria is that both primal residual $\|\mathbf{A}_1\mathbf{x}^k + \mathbf{A}_2\mathbf{y}^k - \overline{\mathbf{b}}\|_\infty$ and dual residual $\|\mathbf{A}_1^T\mathbf{z}^k + \mathbf{c}\|_\infty$ is less than $10^{-3}$. All the LP instances are generated from the basis pursuit, L1 SVM, SICE and NMF problems. The data source and statistics are included in the supplementary material.

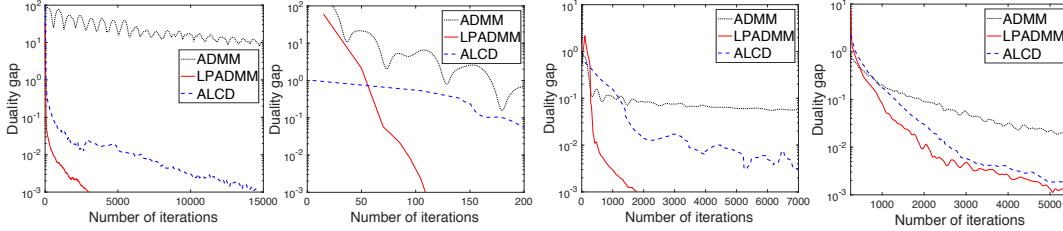

Figure 1: The duality gap versus the number of iterations. From left to right figures are the BP, NMF, the L1 SVM and and the SICE problem.

Table 1: Timing Results for BP, SICE, NMF and L1 SVM Problem (in sec. long means > 60 hours)

| Data | $m$ | $n$ | nnz($\mathbf{A}$) | LPADMM | | ALCD | | ADMM | |
|------|-----|-----|-------------------|--------|------------|------|------------|------|------------|
| | | | | Time | Iterations | Time | Iterations | Time | Iterations |
| bp1 | 17408 | 16384 | 8421376 | **22** | **3155** | 864 | 14534 | long | long |
| bp2 | 34816 | 32768 | 33619968 | **79** | **4657** | 2846 | 19036 | long | long |
| bp3 | 69632 | 65536 | 134348800 | **217** | **6287** | 12862 | 24760 | long | long |
| arcene | 50095 | 30097 | 1151775 | **801** | **15198** | 1978 | 176060 | 21329 | 2035415 |
| real-sim | 176986 | 135072 | 7609186 | **955** | **4274** | 1906 | 18262 | 19697 | 249363 |
| sonar | 80912 | 68224 | 2756832 | **258** | **5446** | 659 | 13789 | 3828 | 151972 |
| colon | 217580 | 161040 | 8439626 | **395** | **216** | 455 | 1288 | 7423 | 83680 |
| w2a | 12048256 | 12146960 | 167299110 | **19630** | **2525** | 45388 | 8492 | long | long |
| news20 | 2785205 | 2498375 | 53625267 | **7765** | **2205** | 9173 | 6174 | long | long |

We first compare the convergence rate of different algorithms in solving the above problems. We use the bp1 for BP problem, data set colon cancer for NMF problem, news20 for L1 SVM problem and real-sim for SICE problem. We set proximity parameter $\rho = 1$. We adopt the relative duality gap as the comparison metric, which is defined as $\|\mathbf{c}^T\mathbf{x}^k + \mathbf{b}^T\mathbf{z}_x^k\|/\|\mathbf{c}^T\mathbf{x}^*\|$, where $\mathbf{x}^*$ is obtained approximately by running our method with a strict stopping condition. In our simulation, one iteration represents $n$ coordinate descent steps for ALCD and LPADMM, and one dual updating step for ADMM. As can be seen in the Fig. 1, our new method exhibits a global and linear convergence rate and matches our theoretical performance bound. Besides, it converges faster than both the ALCD and existing ADMM method, especially in solving the BP and NMF problem. The sensitivity analysis of $\rho$ is listed in Appendix.

We next examine the performance of our algorithm from the perspective of time efficiency (both clocking time and number of iterations). We adopt the dynamic step size rule for ALCD to optimize its performance. Note that, exchanging the role of the primal and dual problem in (3), we can obtain the dual version of both ADMM and ACLD, which can be used to tackle the primal or dual sparse problem. We run both methods and adopt the minimum time. The stopping criterion requires that the primal and dual residual and the relative duality gap is less than $10^{-3}$. The data set bp1,bp2,bp3 is used for basis pursuit problem, news20 is used for L1 SVM problem; arcene, real-sim are used for SICE problem; sonar, colon and w2a are used for NMF problem. Among all experiments, we can observe that our proposed algorithm requires approximately $10\% - 40\%$ iterations and $10\% - 85\%$ time of the ALCD method, and become particularly advantageous for basis pursuit problem ($50\times$ speed up) or ill posed problems such as SICE and NMF problem. In particular, for the basis pursuit problem, the primal iterates $\mathbf{x}^k$ is updated by closed-form expression (15), which can be calculated in $O(n\log(n))$ time by Fast Walsh–Hadamard transform.

## 7 Conclusions

In this paper, we proposed a new variable splitting method to solve the linear programming problem. The theoretical contribution of this work is that we prove that $2-$block ADMM converges globally and linearly when applying to the linear program. The obtained convergence rate has a weak dependence of the problem dimension and is less than the best known result. Compared with the existing LP solvers, our algorithms not only provides a flexibility to exploit the specific structure of constraint matrix $\mathbf{A}$, but also can be naturally combined with the existing acceleration techniques to significantly speed up solving the large-scale machine learning problems. The future work focuses on generalizing our theoretical framework and exhibiting the global linear convergence rate when applying ADMM to solve a convex quadratic program.

**Acknowledgments:** This work is supported by ONR N00014-17-1-2417, N00014-15-1-2166, NSF CNS-1719371 and ARO W911NF-1-0277.

## Footnotes

[1]This technique is motivated by [22].

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
