[Supplementary Material · NIPS_947_Appendix.pdf]

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

$$\mathbf{prox}_{\rho f}(\mathbf{x}) = \arg \min_{\mathbf{u}} f(\mathbf{u}) + \frac{1}{2\rho}\|\mathbf{u} - \mathbf{x}\|^2. \tag{2}$$

A twice differentiable function $f : \mathbb{R}^n \to \mathbb{R}$ has strong convexity parameter $\rho$ if and only if its Hessian satisfies $\nabla^2 f(\mathbf{x}) \succeq \rho\mathbf{I}, \forall\mathbf{x}$. We use $\| \cdot \|$ to denote standard $l_2$ norm for vector or spectral norm for matrix, $\| \cdot \|_1$ to denote the $l_1$ norm and $\| \cdot \|_F$ to denote the Frobenius norm. A twice differentiable function $f : \mathbb{R}^n \to \mathbb{R}$ has the component-wise Lipschitz continuous gradient with constant $L_i$ if and only if $\|\nabla_i f(\mathbf{x}) - \nabla_i f(\mathbf{y})\| \le L_i\|\mathbf{x} - \mathbf{y}\|, \forall\mathbf{x}, \mathbf{y}$. For example, for the quadratic function $F(\mathbf{x}) = \frac{1}{2}\|\mathbf{A}\mathbf{x} - \mathbf{b}\|^2$, the gradient $\nabla F(\mathbf{x}) = \mathbf{A}^T(\mathbf{A}\mathbf{x} - \mathbf{b})$ and the Hessian $\nabla^2 F(\mathbf{x}) = \mathbf{A}^T\mathbf{A}$. Hence the parameter $\rho$ and $L_i$ satisfies (choose $\mathbf{y} = \mathbf{x} + t\mathbf{e}_i$, where $t \in \mathbb{R}$, $\mathbf{e}_i \in \mathbb{R}^n$ is the unit vector), $\mathbf{x}\mathbf{A}^T\mathbf{A}\mathbf{x} \ge \rho\|\mathbf{x}\|^2$ and $t\mathbf{A}_i^T\mathbf{A}\mathbf{e}_i \le L_i|t|, \forall\mathbf{x}, t$. Thus, the $\rho$ is the smallest eigenvalue of $\mathbf{A}^T\mathbf{A}$ and $L_i = \|\mathbf{A}_i\|^2$, where $\mathbf{A}_i$ is $i$th column of the matrix $\mathbf{A}$. The projection operator of point $\mathbf{x}$ into convex set $\mathcal{S}$ is defined as $[\mathbf{x}]_{\mathcal{S}} = \arg\min_{\mathbf{u}\in\mathcal{S}}\|\mathbf{x} - \mathbf{u}\|$. If $\mathcal{S}$ is the non-negative cone, let $[\mathbf{x}]_+ \triangleq [\mathbf{x}]_{\mathcal{S}}$. Let $V_i = [0, \infty)$ for $i \in [n_b]$ and $V_i = \mathbb{R}$ for $i \in [n]\backslash[n_b]$.

### 2.2 Applications

**Basis pursuit problem:** The problem of basis pursuit [4] is a fundamental decoding model in the compressive sensing. It aims at recovering the original signal from the compressed one with preserving the sparsity.

$$\min_{\mathbf{x}\in R^n} \|\mathbf{x}\|_1$$

$$s.t. \quad \mathbf{A}\mathbf{x} = \mathbf{b}, \tag{3}$$

where $\mathbf{A} \in \mathbb{R}^{m \times n}$ is the sensing matrix and $\mathbf{b}$ is the compressed measurement. In practice, the dimension $m \ll n$ and matrix $\mathbf{A}$ are formed by randomly taking a subset of rows from orthonormal transform matrices, such as DCT (discrete cosine transform), DFT (discrete Fourier transform) or DWHT (discrete Walsh-Hadamard transform) matrices.

$l_1-$**regularized SVM:** The problem of $l_1-$regularized support vector machine [2] aims at finding a classifier in the 2-class SVM,

$$\min_{\boldsymbol{\beta}} \sum_{i=1}^{n} \left[ 1 - y_i(\boldsymbol{\beta}^T \mathbf{x}_i) \right]_+ + \lambda \|\boldsymbol{\beta}\|_1, \tag{4}$$

where $(\mathbf{x}_i, y_i)$ are the $i$th training data and label, $\boldsymbol{\beta} \in \mathbb{R}^p$ is the linear classifier, and $\lambda$ is the tuning parameter. It can be generalized to the scenario of multi-classes by replacing the hinge loss term in the objective function of maximizing the distances among different class, i.e., $\boldsymbol{\beta}_1^T \mathbf{x}_i - \boldsymbol{\beta}_2^T \mathbf{x}_i$.

**Sparse Inverse Covariance Matrix Estimation:** This problem aims to find a sparse matrix to approximate the inverse of the covariance matrix $\mathbf{S} \in \mathbb{R}^{p \times p}$. One popular approach [3] is to solve $p$ independent problems of the following form

$$\min_{\beta \in \mathbb{R}^{p-1}} \|\boldsymbol{\beta}\|_1 \quad \text{s.t.} \quad \|\mathbf{S}_{-i,j} - \mathbf{S}_{-i,-i}\boldsymbol{\beta}\|_\infty \le \delta, \tag{5}$$

where $\mathbf{S}_{-i,j}$ is the $j$th column of $\mathbf{S}$ with its $i$th entry removed, $\mathbf{S}_{-i,-i}$ is the submatrix of $\mathbf{S}$ with its $i$th row and column removed, and $\delta$ is predefined approximation threshold. In practice, the covariance matrix $\mathbf{S}$ is dense, but it can be decomposed into the product of two sparse matrices, i.e., $\mathbf{S} = \mathbf{X}\mathbf{X}^T$, which can be exploited by introducing the auxiliary variables in LP.

**Nonnegative Matrix Factorization:** Given $n$ nonnegative $m-$dimensional vectors collected in matrix $\mathbf{M} \in \mathbb{R}_+^{m \times n}$, the NMF determines two nonnegative matrix $\mathbf{W} \in \mathbb{R}_+^{m \times r}$ and $\mathbf{H} \in \mathbb{R}_+^{r \times n}$ such that $\mathbf{M} \approx \mathbf{W}\mathbf{H}$. It is a powerful technique in dimensionality reduction and can be solved in polynomial time when the matrix $\mathbf{M}$ satisfies a separability condition. One of most popular approaches [1, 10] to solve this problem is

$$\min_{\mathbf{X} \in R_+^{n \times n}} \mathbf{p}^T \text{diag}(\mathbf{X})$$
$$s.t. \quad \|\tilde{\mathbf{M}} - \tilde{\mathbf{M}}\mathbf{X}\|_1 \le \lambda\epsilon, X_{ii} \le 1, X_{ij} \le X_{ii}, \forall i,j,$$

where $\mathbf{p}$ is any $n-$dimensional vector with distinct entries, $\tilde{\mathbf{M}}$ is a normalized noisy matrix, $\lambda$ is predefined approximation threshold.

## 2.3 Tail Convergence of the Existing ADMM Method

The existing ADMM in [8] solves the LP (1) by following procedure: in each iteration $k$, go through the following two steps:

1. Primal update: $x_i^{k+1} = \left[ x_i^k + \frac{1}{\|\mathbf{A}_i\|^2} \left( \frac{\mathbf{A}_i^T(\mathbf{b} - \mathbf{A}\mathbf{x}^k)}{q} - \frac{c_i - \mathbf{A}_i^T \mathbf{z}^k}{\lambda} \right) \right]_{V_i}, i = 1, \dots, n.$

2. Dual update: $\mathbf{z}^{k+1} = \mathbf{z}^k - \frac{\lambda}{q}(\mathbf{A}\mathbf{x}^k - \mathbf{b}).$

In the Fig. 1, we plot the solving accuracy versus the number of iterations for solving three kinds of problems: $l_1-$regularized SVM, sparse inverse covariance matrix estimation and nonnegative matrix factorization problem. Here the solving accuracy is defined as the relative accuracy $(\mathbf{c}^T \mathbf{x}^k - \mathbf{c}^T \mathbf{x}^*)/(\mathbf{c}^T \mathbf{x}^*)$, where $\mathbf{x}^*$ is obtained approximately by running our method with a specific topping criterion. The detailed information of date set and LP is listed in TABLE 1.

We can observe that it converges fast in the initial phase, but exhibits a slow and fluctuating convergence when the iterates approach the optimal set. This method originates from a specific splitting method in the standard $2-$block ADMM [11]. To provide some understanding of this phenomenon, we show that this method can be actually recovered by an inexact Uzawa method [12]. The Augmented Lagrangian function of the problem (1) is denoted by $L(\mathbf{x}, \mathbf{z}) = \mathbf{c}^T \mathbf{x} + \frac{\rho}{2}\|\mathbf{A}\mathbf{x} - \mathbf{b} - \mathbf{z}/\rho\|^2$.

Figure 1: The relative duality gap versus the number of iterations. Here one iteration represents one dual updating step in existing ADMM.

Table 1: Data Statistics for Testing Existing ADMM (w8a is used for L1 SVM, rcv1 is used for SICE and a1a is used for NMF problem)

| Data Set | #Samples | #Features | $m$ | $n$ | $nnz(\mathbf{A})$ |
|---|---|---|---|---|---|
| w8a | 49749 | 300 | 50095 | 30097 | 1161572 |
| rcv1 | 15564 | 47236 | 256417 | 161947 | 7339063 |
| a1a | 1605 | 123 | 3150615 | 2958015 | 42006060 |

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

$$\min \quad \mathbf{c}^T \mathbf{x} + \|\mathbf{x}\|_1 \tag{15}$$

$$s.t. \quad \mathbf{A}\mathbf{x} = \mathbf{b}. \tag{16}$$

Transforming to the canonical form (1) will add additional $n$ variables and $2n$ constraints. One important feature in our method is that we can split the objective function by adding group of variable $\mathbf{y}$.

$$\min \quad \mathbf{c}^T\mathbf{x} + \|\mathbf{y}\|_1 \tag{17}$$

$$s.t. \quad \mathbf{A}\mathbf{x} = \mathbf{b}, \mathbf{x} = \mathbf{y}. \tag{18}$$

Similarly, applying the ADMM to problem (17), we can obtain the following two subproblems: the first step is an unconstrained QP,

$$\mathbf{x}^{k+1} = \arg\min_{\mathbf{x}} \mathbf{c}^T\mathbf{x} + (\mathbf{z}^k)^T\mathbf{A}_1\mathbf{x} + \frac{\rho}{2}\|\mathbf{A}_1\mathbf{x} + \mathbf{A}_2\mathbf{y}^k - \overline{\mathbf{b}}\|^2.$$

The second step is $n$ one$-$dimensional shrinkage operations: for each $i$,

$$y_i^{k+1} = \begin{cases} x_i^{k+1} + (z_i^k - 1)/\rho, & \text{if } x_i^{k+1} + z_i^k/\rho \geq 1/\rho \\ x_i^{k+1} + (z_i^k + 1)/\rho, & \text{if } x_i^{k+1} + z_i^k/\rho \leq -1/\rho \\ 0, & \text{otherwise} \end{cases}.$$

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

2. Sample coordinate $i$ with probability proportional to $\sqrt{L_i}$ and update $\mathbf{x}_{t+1}^{k+1} = \mathbf{u}_t - \frac{1}{L_i}\nabla_i F_k(u_t)\mathbf{e}_i$, where $L_i$ is the Lipschitz constant of $\nabla_i F_k(\cdot)$.

3. $\mathbf{v}_{t+1} = \frac{1}{1+\eta\rho}\left(\mathbf{v}_t + \eta\rho\mathbf{u}_t - \frac{\eta}{p_i}\nabla_i F_k(\mathbf{u}_t)\mathbf{e}_i\right).$

Here we drop the superscript $k+1$ of intermediate variables $\mathbf{u}, \mathbf{v}$ in each inner iteration. In the above procedure, $\eta = \frac{1}{\tau S^2}, \tau = \frac{2}{1+\sqrt{4S^2/\rho+1}}, S = \sum_{i=1}^n \sqrt{\|\mathbf{A}_i\|^2 + 1}$. This choice of parameter is based on letting $\beta = 0$ in the original ACDM [22] and the estimation of component-wise Lipschitz constant $L_i = \|\mathbf{A}_i\|^2 + 1$ and strongly convexity parameter $\sigma \geq \rho$ of problem (11). The above procedure can be further written as

$$\mathbf{u}_{t+1} = \tau\mathbf{v}_{t+1} + (1-\tau)\mathbf{x}_{t+1}^{k+1}$$
$$= \frac{1+\eta\rho-\tau}{1+\eta\rho}\mathbf{u}^t + \frac{\tau}{1+\eta\rho}\mathbf{v}^t - \left[\frac{\eta\tau}{p_i(1+\eta\rho)} + \frac{1-\tau}{L_i}\right]\nabla_i F_k(\mathbf{u}_t)\mathbf{e}_i.$$

Combining this formula with the third step of ACDM, we can write each iteration as following matrix iteration:

$$\begin{bmatrix}\mathbf{u}_{t+1}^T \\ \mathbf{v}_{t+1}^T\end{bmatrix} = \mathbf{M}\begin{bmatrix}\mathbf{u}_t^T \\ \mathbf{v}_t^T\end{bmatrix} - \mathbf{s}_t^i. \tag{25}$$

The matrix $\mathbf{M}$ and vector $\mathbf{s}_t^i$ are defined as

$$\mathbf{M} = \begin{bmatrix}1-\alpha_v & \alpha_v \\ \beta_u & 1-\beta_u\end{bmatrix} \text{ with } \begin{bmatrix}\alpha_v \\ \beta_u\end{bmatrix} = \begin{bmatrix}\frac{\tau}{1+\eta\rho} \\ \frac{\eta\rho}{1+\eta\rho}\end{bmatrix} \text{ and } \mathbf{s}_t^i = \begin{bmatrix}\left(\frac{\eta\tau}{p_i(1+\eta\rho)} + \frac{1-\tau}{L_i}\right)\nabla_i F_k(\mathbf{u}_t)\mathbf{e}_i^T \\ \frac{\eta}{p_i(1+\eta\rho)}\nabla_i F_k(\mathbf{u}_t)\mathbf{e}_i^T\end{bmatrix}, \tag{26}$$

Therefore, we can implement ACDM in each iteration as

$$\begin{bmatrix}\mathbf{u}_{t+1}^T \\ \mathbf{v}_{t+1}^T\end{bmatrix} = \mathbf{M}_{t+1}\begin{bmatrix}\mathbf{u}^T \\ \mathbf{v}^T\end{bmatrix}, \tag{27}$$

and update $\mathbf{u}, \mathbf{v}$ and matrix $\mathbf{M}_t$ by

$$\mathbf{M}_{t+1} = \mathbf{M} \cdot \mathbf{M}_t \text{ and } \begin{bmatrix}\mathbf{u}^T \\ \mathbf{v}^T\end{bmatrix} := \begin{bmatrix}\mathbf{u}^T \\ \mathbf{v}^T\end{bmatrix} - \mathbf{M}_{t+1}^{-1}\mathbf{s}_t^i. \tag{28}$$

The rest is to calculate the partial derivative

$$\nabla_i F_k(\mathbf{u}_t) = \rho(\mathbf{A}_i)^T\mathbf{A}\mathbf{u}_t + \rho u_{t,i} + \mathbf{A}_{1,i}^T[\mathbf{z}^k + \rho(\mathbf{A}_2\mathbf{y}^k - \overline{\mathbf{b}})] + c_i,$$

where $\mathbf{A}_{1,i}$ is $i$th column of matrix $\mathbf{A}_1$. Utilizing auxillary variable $\overline{\mathbf{u}}_t$ and $\overline{\mathbf{v}}_t$ to represent $\mathbf{A}\mathbf{u}_t$ and $\mathbf{A}\mathbf{v}_t$ and multiplying (25) by matrix $\mathbf{A}$, we have

$$\begin{bmatrix}(\mathbf{A}\mathbf{u}_{t+1})^T \\ (\mathbf{A}\mathbf{v}_{t+1})^T\end{bmatrix} = \mathbf{M}\begin{bmatrix}(\mathbf{A}\mathbf{u}_t)^T \\ (\mathbf{A}\mathbf{v}_t)^T\end{bmatrix} + \mathbf{s}_t^i\mathbf{A}^T \tag{29}$$

$$\Longleftrightarrow \begin{bmatrix}\overline{\mathbf{u}}_{t+1}^T \\ \overline{\mathbf{v}}_{t+1}^T\end{bmatrix} = \mathbf{M}\begin{bmatrix}\overline{\mathbf{u}}_t^T \\ \overline{\mathbf{v}}_t^T\end{bmatrix} + \mathbf{s}_t^i\mathbf{A}^T. \tag{30}$$

Therefore, to implement ACDM in each iteration, we can just maintain vectors $\overline{\mathbf{u}}$ and $\overline{\mathbf{v}}$ such that

$$\begin{bmatrix}\overline{\mathbf{u}}_{t+1}^T \\ \overline{\mathbf{v}}_{t+1}^T\end{bmatrix} = \mathbf{M}\begin{bmatrix}\overline{\mathbf{u}}^T \\ \overline{\mathbf{v}}^T.\end{bmatrix} \tag{31}$$

With this representation, each update step can be implemented by

$$\mathbf{M}_{t+1} = \mathbf{M} \cdot \mathbf{M}_t \text{ and } \begin{bmatrix}\overline{\mathbf{u}}^T \\ \overline{\mathbf{v}}^T\end{bmatrix} := \begin{bmatrix}\overline{\mathbf{u}}^T \\ \overline{\mathbf{v}}^T\end{bmatrix} - \mathbf{M}_{t+1}^{-1}\mathbf{s}_t^i\mathbf{A}^T. \tag{32}$$

**Lemma 6.** (Inner complexity) *In each iteration of Algorithm 2, if the current picked coordinate is $i$, the update can be finished in $O(nnz(\mathbf{A}_i))$ time, moreover, to guarantee that $F_k(\mathbf{x}^{k+1}) - \min_{\mathbf{x}} F_k(\mathbf{x}) \leq \epsilon_k$ with probability $1 - p$, it suffices to run Algorithm 2 for number of iterations*

$$T_k \geq O(1) \cdot \sum_{i=1}^n \|\mathbf{A}_i\| \log\left(\frac{D_0^k}{\epsilon_k p}\right), \quad D_0^k = \|\overline{F}_k(\mathbf{u}_0) - \min_{\mathbf{x}} \overline{F}_k(\mathbf{x})\|. \tag{33}$$

Table 2: Data Statistics for Experiments in Basis Pursuit Problem

| Data Set | Signal dimension | Measurements dimension |
|----------|------------------|------------------------|
| bp1 | 8192 | 1024 |
| bp2 | 16384 | 2048 |
| bp3 | 32768 | 4096 |

Table 3: Data Statistics for Experiments in L1 SVM, SICE and NMF Problem

| Data Set | #Samples | #Features | nnz |
|----------|----------|-----------|-----|
| news20 | 15935 | 62061 | 1272569 |
| real-sim | 72309 | 20958 | 3709083 |
| arcene | 900 | 10000 | 540941 |
| colon | 62 | 2000 | 124000 |
| sonar | 208 | 60 | 12479 |
| w2a | 3470 | 300 | 40373 |

The above iteration complexity is obtained by choosing parameter $\beta = 0$ in [22] and utilizing the Theorem 1 in [24] to transform the convergence in expectation to the form of probability.

**Theorem 2.** (Overall complexity) *Denote $\mathbf{z}^k$ as the dual iterates produced by Algorithm 1. To guarantee that there exists an optimal solution $\mathbf{z}^*$ such that $\|\mathbf{z}^k - \mathbf{z}^*\| \leq \epsilon$ with probability $1 - p$, it suffices to run Algorithm 1 for*

$$k \geq 2\gamma^2 \log(2D_0/\epsilon) \tag{34}$$

*outer iterations and solve each sub-problem (11) for the number of inner iterations*

$$T \geq O(1) \cdot \sum_{i=1}^{n} \|\mathbf{A}_i\| \log \left( \frac{\rho(D_0^k)^{\frac{1}{3}} \gamma^2}{\epsilon^{\frac{2}{3}} p^{\frac{1}{3}}} \log \left( \frac{2D_0}{\epsilon} \right) \right). \tag{35}$$

The results for the primal iterates $\mathbf{x}^k$ and $\mathbf{y}^k$ are similar. In the existing ADMM [8], each primal and dual update only requires $O(nnz(\mathbf{A}))$ time to solve. The complexity of this method is

$$O(a_m \mu^2 (a_m R_x + d_m R_z)^2 (\sqrt{mn} + \|\mathbf{A}\|_F)^2 nnz(\mathbf{A}) \log(1/\epsilon)),$$

where $a_m = \max_i \|\mathbf{A}_i\|$, $d_m$ is the largest number of non-zero elements of each row of matrix $\mathbf{A}$, and $\mu$ is the Hoffman constant depends on the optimal solution set of LP. Based on Theorem 2, an estimation of the worst-case complexity of Algorithm 1 is

$$O(a_m \theta_{S^*}^2 (R_x \|\mathbf{A}\| + R_z)^2 nnz(\mathbf{A}) \log^2(1/\epsilon)).$$

Remark that our method has a weak dependence on the problem dimension compared with the existing ADMM. Since the Frobenius norm of a matrix satisfies $\|\mathbf{A}\|_2 \leq \|\mathbf{A}\|_F$, our method is faster than the one in [8].

## 6 Numerical Results

In this section, we examine the performance of our algorithm and compare it with the state-of-art of algorithms developed for solving the LP. The first is the existing ADMM in [8]. The second is the ALCD method in [9], which is reported to be the current fastest first-order LP solver. They have shown that this algorithm can significantly speed up solving several important machine learning problems compared with the Simplex and IPM. We name our Algorithm 1 as LPADMM. In the experiments, we require that the accuracy of subproblem solver $\epsilon_k = 10^{-3}$ and the stopping criteria is that both primal residual $\|\mathbf{A}_1 \mathbf{x}^k + \mathbf{A}_2 \mathbf{y}^k - \overline{\mathbf{b}}\|_\infty$ and dual residual $\|\mathbf{A}_1^T \mathbf{z}^k + \mathbf{c}\|_\infty$ is less than $10^{-3}$. All the LP instances are generated from the basis pursuit, L1 SVM, SICE and NMF problems. The data source and statistics are listed in TABLE 2 and TABLE 3.

For the basis pursuit problem, we adopt the following popular signal generation model [4]. In particular, the target signal $\mathbf{x} \in \mathbb{R}^n$ is set to

$$x_i = \mathbf{1}\{i \in \Lambda\} \Theta_i^{(1)} 10^{2\Theta_i^{(2)}}, \tag{36}$$

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

Under the same stopping criterion, we vary the Augmented Lagrangian parameter $\rho$ and see how it influences the required number of iterations to obtain a given accuracy solution. We also run 100 times with the same $\rho$ for each data set to avoid random noise. We observe that, when the parameter $\rho$ increases 100 times (from 1 to 100), the number of iterations decreases or increases roughly 20% compared with the existing results in TABLE 1 (section of numerical results). Moreover, even

when the $\rho$ is drastically increased from 1 to 100, the largest number of iterations and clocking time produced by our algorithm are still much less than the smallest one produced by other algorithms.

# 7 Conclusions

In this paper, we proposed a new variable splitting method to solve the linear programming problem. The theoretical contribution of this work is that we prove that $2-$block ADMM converges globally and linearly when applying to the linear program. The obtained convergence rate has a weak dependence of the problem dimension and is less than the best known result. Compared with the existing LP solvers, our algorithms not only provides a flexibility to exploit the specific structure of constraint matrix $\mathbf{A}$, but also can be naturally combined with the existing acceleration techniques to significantly speed up solving the large-scale machine learning problems. The future work focuses on generalizing our theoretical framework and exhibiting the global linear convergence rate when applying ADMM to solve a convex quadratic program.

**Acknowledgments:** This work is supported by ONR N00014-17-1-2417, N00014-15-1-2166, NSF CNS-1719371 and ARO W911NF-1-0277.

## Footnotes

[1]This technique is motivated by [23].

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

## A    Proof of Lemma 1

The proof of Lemma 1 is tailored from [11]. Several intermediate results in this proof will be useful in the derivation of other lemmas. Thus, we provide this modification for completeness. We first write the three steps of ADMM as,

$$\mathbf{x}^{k+1} = \arg\min_{\mathbf{x}} f(\mathbf{x}) + (\mathbf{z}^k)^T \mathbf{A}_1 \mathbf{x} + \frac{\rho}{2} \|\mathbf{A}_1 \mathbf{x} + \mathbf{A}_2 \mathbf{y}^k - \overline{\mathbf{b}}\|^2, \tag{37}$$

$$\mathbf{y}^{k+1} = \arg\min_{\mathbf{y}} g(\mathbf{y}) + (\mathbf{z}^k)^T \mathbf{A}_2 \mathbf{y} + \frac{\rho}{2} \|\mathbf{A}_1 \mathbf{x}^{k+1} + \mathbf{A}_2 \mathbf{y} - \overline{\mathbf{b}}\|^2, \tag{38}$$

$$\mathbf{z}^{k+1} = \mathbf{z}^k + \rho(\mathbf{A}_1 \mathbf{x}^{k+1} + \mathbf{A}_2 \mathbf{y}^{k+1} - \overline{\mathbf{b}}). \tag{39}$$

We will prove (37)-(39) are equivalent to the following proximal operations.

$$\mathbf{w}^{k+1} = \mathbf{prox}_{\rho \overline{f}}(\mathbf{z}^k + \rho \mathbf{A}_2 \mathbf{y}^k), \tag{40}$$

$$\mathbf{z}^{k+1} = \mathbf{prox}_{\rho \overline{g}}(\mathbf{w}^{k+1} - \rho \mathbf{A}_2 \mathbf{y}^k), \tag{41}$$

$$\rho \mathbf{A}_2 \mathbf{y}^{k+1} = \rho \mathbf{A}_2 \mathbf{y}^k + \mathbf{z}^{k+1} - \mathbf{w}^{k+1}, \tag{42}$$

where $\mathbf{w}^{k+1} = \mathbf{z}^k + \rho(\mathbf{A}_1 \mathbf{x}^{k+1} + \mathbf{A}_2 \mathbf{y}^k - \overline{\mathbf{b}})$ are the dual variables in the optimization problem (11). Functions $f$ and $g$ is defined as

$$\overline{f}(\mathbf{u}) \triangleq \overline{\mathbf{b}}^T \mathbf{u} + f^*(-\mathbf{A}_1^T \mathbf{x}) \quad \text{and} \quad \overline{g}(\mathbf{u}) \triangleq g^*(-\mathbf{A}_2^T \mathbf{x}), \tag{43}$$

where $f^*$ and $g^*$ is the convex conjugate of function $f$ and $g$, defined as $f^*(\mathbf{y}) = \sup_{\mathbf{x}} \mathbf{y}^T \mathbf{x} - f(\mathbf{x})$ and $g^*(\mathbf{y}) = \sup_{\mathbf{x}} \mathbf{y}^T \mathbf{x} - g(\mathbf{x})$.

**Claim** 1: equivalence of first step (37) $\iff$ (40)

$$\mathbf{x}^{k+1} = \arg\min_{\mathbf{x}} f(\mathbf{x}) + (\mathbf{z}^k)^T \mathbf{A}_1\mathbf{x} + \frac{\rho}{2}\|\mathbf{A}_1\mathbf{x} + \mathbf{A}_2\mathbf{y}^k - \overline{\mathbf{b}}\|^2$$

$$\stackrel{(a)}{\iff} \mathbf{0} \in \partial f(\mathbf{x}^{k+1}) + \mathbf{A}_1^T\mathbf{z}^k + \rho\mathbf{A}_1^T(\mathbf{A}_1\mathbf{x}^{k+1} + \mathbf{A}_2\mathbf{y}^k - \overline{\mathbf{b}}) \qquad (44)$$

$$\stackrel{(b)}{\iff} -\mathbf{A}_1^T\mathbf{w}^{k+1} \in \partial f(\mathbf{x}^{k+1})$$

$$\stackrel{(c)}{\iff} \mathbf{x}^{k+1} \in \partial f^*(-\mathbf{A}_1^T\mathbf{w}^{k+1})$$

$$\stackrel{(d)}{\iff} -\mathbf{A}_1\mathbf{x}^{k+1} \in -\mathbf{A}_1\partial f^*(-\mathbf{A}_1^T\mathbf{w}^{k+1})$$

$$\stackrel{(e)}{\iff} \mathbf{0} \in \overline{\mathbf{b}} - \mathbf{A}_1\partial f^*(-\mathbf{A}_1^T\mathbf{w}^{k+1}) + \frac{1}{\rho}(\mathbf{w}^{k+1} - \mathbf{z}^k - \rho\mathbf{A}_2\mathbf{y}^k)$$

$$\stackrel{(f)}{\iff} \mathbf{w}^{k+1} = \mathbf{prox}_{\rho\overline{f}}(\mathbf{z}^k + \rho\mathbf{A}_2\mathbf{y}^k).$$

The above, (a) is based on the first-order optimality condition of unconstrained optimization, (b) utilizes the definition of $\mathbf{w}^{k+1}$, (c) is based on the fact that function $f$ is closed and convex, (d) utilizes the fact that matrix $\mathbf{A}_1$ has full column rank, since $\mathbf{A}_1 = [\mathbf{A}; \mathbf{I}]$ contains the identity matrix in its column space. (e) utilizes the definition of $\mathbf{w}^{k+1}$ again. (f) is based on the first-order optimality condition of proximal operator.

**Claim** 2: equivalence of second step (38) $\iff$ (41)

$$\mathbf{y}^{k+1} = \arg\min_{\mathbf{y}} g(\mathbf{y}) + (\mathbf{z}^k)^T \mathbf{A}_2\mathbf{y} + \frac{\rho}{2}\|\mathbf{A}_1\mathbf{x}^{k+1} + \mathbf{A}_2\mathbf{y} - \overline{\mathbf{b}}\|^2$$

$$\stackrel{(a)}{\iff} \mathbf{0} \in \partial g(\mathbf{y}^{k+1}) + \mathbf{A}_2^T\mathbf{z}^k + \rho\mathbf{A}_2^T(\mathbf{A}_1\mathbf{x}^{k+1} + \mathbf{A}_2\mathbf{y}^{k+1} - \overline{\mathbf{b}})$$

$$\stackrel{(b)}{\iff} -\mathbf{A}_2^T\mathbf{z}^{k+1} \in \partial g(\mathbf{y}^{k+1})$$

$$\stackrel{(c)}{\iff} \mathbf{y}^{k+1} \in \partial g^*(-\mathbf{A}_2^T\mathbf{z}^{k+1}) \qquad (45)$$

$$\stackrel{(d)}{\iff} -\mathbf{A}_2\mathbf{y}^{k+1} \in -\mathbf{A}_2\partial g^*(-\mathbf{A}_2^T\mathbf{z}^{k+1})$$

$$\stackrel{(e)}{\iff} \mathbf{0} \in -\mathbf{A}_2\partial g^*(-\mathbf{A}_2^T\mathbf{z}^{k+1}) + \frac{1}{\rho}(\mathbf{z}^{k+1} - \mathbf{w}^{k+1} + \rho\mathbf{A}_2\mathbf{y}^k)$$

$$\stackrel{(f)}{\iff} \mathbf{z}^{k+1} = \mathbf{prox}_{\rho\overline{g}}(\mathbf{w}^{k+1} - \rho\mathbf{A}_2\mathbf{y}^k).$$

The above, (a) is based on the first-order optimality condition of unconstrained optimization, (b) utilizes the definition of dual iterates $\mathbf{z}^{k+1}$, (c) is based on the fact that function $g$ is closed and convex. (d) utilizes the fact that matrix $\mathbf{A}_2$ has full column rank, since $\mathbf{A}_1 = [\mathbf{0}; -\mathbf{I}]$ contains the identity matrix in its column space. (e) utilizes the definition of $\mathbf{z}^{k+1}$ again. (f) is based on the first-order optimality condition of proximal operator.

**Claim** 3: equivalence between (39) $\iff$ (42)

$$\rho\mathbf{A}_2\mathbf{y}^k + \mathbf{z}^{k+1} - \mathbf{w}^{k+1}$$

$$\stackrel{(a)}{=} \rho\mathbf{A}_2\mathbf{y}^k + [\mathbf{z}^k + \rho(\mathbf{A}_1\mathbf{x}^{k+1} + \mathbf{A}_2\mathbf{y}^{k+1} - \overline{\mathbf{b}})] - [\mathbf{z}^k + \rho(\mathbf{A}_1\mathbf{x}^{k+1} + \mathbf{A}_2\mathbf{y}^k - \overline{\mathbf{b}})]$$

$$= \rho\mathbf{A}_2\mathbf{y}^{k+1}.$$

The above, (a) is based on the definition of $\mathbf{w}^{k+1}$ and $\mathbf{z}^{k+1}$.

Combining claim 1-3, we arrive at the desired equivalence between ADMM and the proximal operations. Based on the definition of iterates $\mathbf{p}^k = \mathbf{z}^k - \rho\mathbf{A}_2\mathbf{y}^k$, the proximal operations (40)-(42) are equivalent to

$$\mathbf{w}^{k+1} = \mathbf{prox}_{\rho\overline{f}}(2\mathbf{z}^k - \mathbf{p}^k), \qquad (46)$$

$$\mathbf{z}^{k+1} = \mathbf{prox}_{\rho\overline{g}}(\mathbf{w}^{k+1} - \mathbf{z}^k + \mathbf{p}^k), \qquad (47)$$

$$\mathbf{p}^{k+1} = \mathbf{p}^k + \mathbf{w}^{k+1} - \mathbf{z}^k. \qquad (48)$$

It can be further simplified as

$$\mathbf{p}^{k+1} = \mathbf{p}^k + \mathbf{prox}_{\rho\overline{f}}(2\mathbf{z}^k - \mathbf{p}^k) - \mathbf{z}^k, \tag{49}$$

$$\mathbf{z}^{k+1} = \mathbf{prox}_{\rho\overline{g}}(\mathbf{p}^{k+1}). \tag{50}$$

Start at $\mathbf{p}^0$ and renumber the iterates, we have

$$\mathbf{z}^{k+1} = \mathbf{prox}_{\rho\overline{g}}(\mathbf{p}^k), \tag{51}$$

$$\mathbf{p}^{k+1} = \mathbf{p}^k + \mathbf{prox}_{\rho\overline{f}}(2\mathbf{z}^{k+1} - \mathbf{p}^k) - \mathbf{z}^{k+1}, \tag{52}$$

which is the classic Douglas-Rachford splitting method [11]. We can further write it as following iteration of proximal operator.

$$\mathbf{p}^{k+1} = T(\mathbf{p}^k), \tag{53}$$

where the operator $T$ is defined as

$$T(\mathbf{x}) = \mathbf{x} + \mathbf{prox}_{\rho\overline{f}}(2\mathbf{prox}_{\rho\overline{g}}(\mathbf{x}) - \mathbf{x}) - \mathbf{prox}_{\rho\overline{g}}(\mathbf{x}). \tag{54}$$

**Claim** 4: Let $G(\mathbf{x}) = \mathbf{x} - T(\mathbf{x})$, then $G$ is firmly non-expansive.

Based on the fact that proximal operator $\mathbf{prox}_{\rho\overline{f}}(\cdot)$ and $\mathbf{prox}_{\rho\overline{g}}$ are firmly non-expansive, we have $\forall \mathbf{p}, \mathbf{p}' \in \mathbb{R}^{m+n}$,

$$\langle \mathbf{prox}_{\rho\overline{g}}(\mathbf{p}) - \mathbf{prox}_{\rho\overline{g}}(\mathbf{p}'), \mathbf{p} - \mathbf{p}' \rangle \geq \|\mathbf{prox}_{\rho\overline{g}}(\mathbf{p}) - \mathbf{prox}_{\rho\overline{g}}(\mathbf{p}')\|^2,$$

$$\langle \mathbf{prox}_{\rho\overline{f}}(2\mathbf{prox}_{\rho\overline{g}}(\mathbf{p}) - \mathbf{p}) - \mathbf{prox}_{\rho\overline{f}}(2\mathbf{prox}_{\rho\overline{g}}(\mathbf{p}') - \mathbf{p}'), (2\mathbf{prox}_{\rho\overline{g}}(\mathbf{p}) - \mathbf{p}) - (2\mathbf{prox}_{\rho\overline{g}}(\mathbf{p}') - \mathbf{p}') \rangle$$

$$\geq \|\mathbf{prox}_{\rho\overline{f}}(2\mathbf{prox}_{\rho\overline{g}}(\mathbf{p}) - \mathbf{p}) - \mathbf{prox}_{\rho\overline{f}}(2\mathbf{prox}_{\rho\overline{g}}(\mathbf{p}') - \mathbf{p}')\|^2.$$

Summing the above two inequalities together and rearranging the terms in both sides, we have

$$\langle G(\mathbf{p}) - G(\mathbf{p}'), \mathbf{p} - \mathbf{p}' \rangle \geq \|G(\mathbf{p}) - G(\mathbf{p}')\|^2, \tag{55}$$

which implies the desired result.

Let $\mathbf{p}^*$ be any point satisfying $T(\mathbf{p}^*) = \mathbf{p}^*$, we have

$$\begin{aligned}
\|\mathbf{p}^{k+1} - \mathbf{p}^*\|^2 &= \|\mathbf{p}^{k+1} - \mathbf{p}^k + \mathbf{p}^k - \mathbf{p}^*\|^2 \\
&= \|\mathbf{p}^k - \mathbf{p}^*\|^2 + \|\mathbf{p}^{k+1} - \mathbf{p}^k\|^2 + 2\langle \mathbf{p}^{k+1} - \mathbf{p}^k, \mathbf{p}^k - \mathbf{p}^* \rangle \\
&\stackrel{(a)}{=} \|\mathbf{p}^k - \mathbf{p}^*\|^2 + \|G(\mathbf{p}^k)\|^2 - 2\langle G(\mathbf{p}^k) - G(\mathbf{p}^*), \mathbf{p}^k - \mathbf{p}^* \rangle \\
&\stackrel{(b)}{\leq} \|\mathbf{p}^k - \mathbf{p}^*\|^2 + \|G(\mathbf{p}^k)\|^2 - 2\|G(\mathbf{p}^k) - G(\mathbf{p}^*)\|^2 \\
&= \|\mathbf{p}^k - \mathbf{p}^*\|^2 - \|\mathbf{p}^k - \mathbf{p}^{k+1}\|^2. 
\end{aligned} \tag{56}$$

The above, (a) is based on the definition of $G(\cdot)$ and $\mathbf{p}^*$ is the zero point of operator $G(\cdot)$, (b) is based on the Claim 4 that the operator $G(\cdot)$ is non-firmly expansive. Since $\mathbf{p}^*$ is any zero point of operator $G(\cdot)$, let $\mathbf{p}^* = [\mathbf{p}^k]_{G^*}$, where $G^* = \{\mathbf{p}^*|T(\mathbf{p}^*) = \mathbf{p}^*\}$, we have

$$\|\mathbf{p}^{k+1} - [\mathbf{p}^{k+1}]_{G^*}\|^2 \leq \|\mathbf{p}^{k+1} - [\mathbf{p}^k]_{G^*}\|^2 \leq \|\mathbf{p}^k - [\mathbf{p}^k]_{G^*}\|^2 - \|\mathbf{p}^k - \mathbf{p}^{k+1}\|^2. \tag{57}$$

The convergence and boundedness of $\mathbf{x}^k, \mathbf{y}^k, \mathbf{z}^k$ can be obtained from the above inequality. More details can be seen in Theorem 2 in [8]. Thus, the lemma follows.

# B   Proof of Lemma 2

Let's first prove the "if" direction. Suppose that $(\mathbf{x}^*, \mathbf{y}^*, \mathbf{z}^*)$ satisfy the condition (i)-(iv). The condition (i)-(iii) implies that the $(\mathbf{x}^*, \mathbf{y}^*, \mathbf{z}^*)$ satisfies the primal feasibility: $A\mathbf{x}^* = \mathbf{b}$, $\mathbf{x}^* = \mathbf{y}^*$, $y_i^* \geq 0, i \in [n_b]$; dual feasibility: $-\mathbf{A}^T \mathbf{z}_x^* - \mathbf{z}_y^* = \mathbf{c}$, $z_{y,i}^* \leq 0, i \in [n_b], z_{y,i}^* = 0, i \in [n] \backslash [n_b]$. Based on the weak duality theorem, for any other primal feasible variables $\mathbf{x}, \mathbf{y}$ and dual feasible variables $\mathbf{z}_x, \mathbf{z}_y$,

$$\mathbf{c}^T\mathbf{x} \geq -\mathbf{b}^T\mathbf{z}_{\mathbf{x}}^* \quad \text{and} \quad \mathbf{c}^T\mathbf{x}^* \geq -\mathbf{b}^T\mathbf{z}_x.$$

Then, combining this result with the condition of (iv), we have
$$\mathbf{c}^T\mathbf{x} \geq \mathbf{c}^T\mathbf{x}^* \quad \text{and} \quad \mathbf{b}^T\mathbf{z}_x \geq \mathbf{b}^T\mathbf{z}_\mathbf{x}^*, \tag{58}$$
which implies the $(\mathbf{x}^*, \mathbf{y}^*, \mathbf{z}^*)$ are optimal primal and dual solutions of the LP (7).

We then prove the "only if" direction. Suppose that the $(\mathbf{x}^*, \mathbf{y}^*, \mathbf{z}^*)$ are optimal primal and dual solutions of the LP (7). By the strong duality theorem of LP, we arrive the last condition (iv). Based on the KKT condition, the primal feasibility implies that $\mathbf{A}\mathbf{x}^* = \mathbf{b}, \mathbf{x}^* = \mathbf{y}^*$ and $y_i^* \geq 0, i \in [n_b]$; the constraints on the dual variable implies that $z_{y,i}^* \leq 0, i \in [n_b]$; the Lagrangian of LP: $\mathbf{c}^T\mathbf{x} + g(\mathbf{y}) + \mathbf{z}_x^T(\mathbf{A}\mathbf{x} - \mathbf{b}) + \mathbf{z}_y^T(\mathbf{x} - \mathbf{y})$ with respect to $\mathbf{x}, \mathbf{y}$ vanishes implies that $-\mathbf{A}^T\mathbf{z}_x^* - \mathbf{z}_y^* = \mathbf{c}$ and $z_{y,i}^* = 0, i \in [n]\backslash[n_b]$.

## C   Proof of Lemma 4

We first show that $\mathbf{p}^{k+1} - \mathbf{p}^k = \rho(\mathbf{A}_1\mathbf{x}^{k+1} + \mathbf{A}_2\mathbf{y}^k - \overline{\mathbf{b}})$. Based on the definition of $\mathbf{p}_k = \mathbf{z}^k - \rho\mathbf{A}_2\mathbf{y}^k$, we have
$$\mathbf{p}^{k+1} - \mathbf{p}^k = (\mathbf{z}^{k+1} - \rho\mathbf{A}_2\mathbf{y}^{k+1}) - (\mathbf{z}^k - \rho\mathbf{A}_2\mathbf{y}^k)$$
$$\overset{(a)}{=} \rho(\mathbf{A}_1\mathbf{x}^{k+1} + \mathbf{A}_2\mathbf{y}^k - \overline{\mathbf{b}}). \tag{59}$$
The above, (a) is based on the dual updating step of ADMM.

Second, we show $\mathbf{c} + \mathbf{A}_1^T\mathbf{z}^k = \mathbf{A}_1^T(\mathbf{p}^k - \mathbf{p}^{k+1})$. Based on the first-order optimality condition of the first step of ADMM (44) and the fact that $\partial f(\mathbf{x}^{k+1}) = \{\mathbf{c}\}$, we have
$$\mathbf{c} + \mathbf{A}_1^T\mathbf{z}^k + \rho\mathbf{A}_1^T(\mathbf{A}_1\mathbf{x}^{k+1} + \mathbf{A}_2\mathbf{y}^k - \overline{\mathbf{b}}) = \mathbf{0}. \tag{60}$$
Utilizing the result of (59), we can obtain
$$\mathbf{c} + \mathbf{A}_1^T\mathbf{z}^k + \mathbf{A}_1^T(\mathbf{p}^{k+1} - \mathbf{p}^k) = \mathbf{0}. \tag{61}$$

Third, we show that $\mathbf{c}^T\mathbf{x}^{k+1} + \mathbf{b}^T\mathbf{z}_x^k = \langle \mathbf{A}_1\mathbf{x}^{k+1} - \mathbf{z}^k/\rho, \mathbf{p}^k - \mathbf{p}^{k+1} \rangle$. Based on the result of (59) and (61), we have
$$\mathbf{A}_1\mathbf{x}^{k+1} + \mathbf{A}_2\mathbf{y}^k - \overline{\mathbf{b}} = (\mathbf{p}^{k+1} - \mathbf{p}^k)/\rho, \tag{62}$$
$$\mathbf{c} + \mathbf{A}_1^T\mathbf{z}^k = \mathbf{A}_1^T(\mathbf{p}^k - \mathbf{p}^{k+1}). \tag{63}$$
Further, based on the second and third steps of ADMM, for those bounded variables, we have ,
$$y_i^{k+1} = [x_i^{k+1} + z_{y,i}^k/\rho]_+ = \max\{0, x_i^{k+1} + z_{y,i}^k/\rho\} \geq 0, i \in [n_b], \tag{64}$$
$$z_{y,i}^{k+1} = z_{y,i}^k + \rho(x_i^{k+1} - y_i^{k+1}) = \min\{z_{y,i}^k + \rho x_i^{k+1}, 0\} \leq 0, i \in [n_b]. \tag{65}$$
For the free variables, we have
$$y_i^{k+1} = x_i^{k+1} + z_{y,i}^k/\rho \quad \text{and} \quad z_{y,i}^{k+1} = z_{y,i}^k + \rho(x_i^{k+1} - y_i^{k+1}) = 0, i \in [n]\backslash[n_b]. \tag{66}$$
Thus, we can obtain,
$$y_i^{k+1} \cdot z_{y,i}^{k+1} = \max\{0, x_i^{k+1} + z_{y,i}^k/\rho\} \cdot \min\{z_{y,i}^k + \rho x_i^{k+1}, 0\} = 0, \quad i \in [n_b], \tag{67}$$
$$y_i^{k+1} \cdot z_{y,i}^{k+1} = (x_i^{k+1} + z_{y,i}^k/\rho) \cdot 0 = 0, \quad i \in [n]\backslash[n_b]. \tag{68}$$
The equations (62)-(68) imply that the pair of iterates $(\mathbf{x}^{k+1}, \mathbf{y}^k)$ and $z^k$ satisfies the complementary slackness of the following perturbed primal and dual LPs. Thus the iterates $(\mathbf{x}^{k+1}, \mathbf{y}^k)$ and $\mathbf{z}^k$

Approximate primal LP

$\min \quad \langle \mathbf{c} - \mathbf{A}_1^T(\mathbf{p}^k - \mathbf{p}^{k+1}), \mathbf{x} \rangle$

$s.t. \quad \mathbf{A}_1\mathbf{x} + \mathbf{A}_2\mathbf{y} = \overline{\mathbf{b}} - (\mathbf{p}^{k+1} - \mathbf{p}^k)/\rho$
$\qquad y_i \geq 0, i \in [n_b].$

Approximate dual LP

$\min \quad \langle \overline{\mathbf{b}} - (\mathbf{p}^{k+1} - \mathbf{p}^k)/\rho, \mathbf{z} \rangle$

$s.t. \quad -\mathbf{A}_1^T\mathbf{z} = \mathbf{c} - \mathbf{A}_1^T(\mathbf{p}^k - \mathbf{p}^{k+1}),$
$\qquad z_{y,i} \leq 0, i \in [n_b], z_{y,i} = 0, i \in [n]\backslash[n_b].$

constitute the optimal primal and dual solutions of the above LPs. According to the strong duality, we have
$$\langle \mathbf{c} - \mathbf{A}_1^T(\mathbf{p}^k - \mathbf{p}^{k+1}), \mathbf{x}^{k+1} \rangle + \langle \overline{\mathbf{b}} - (\mathbf{p}^{k+1} - \mathbf{p}^k)/\rho, \mathbf{z}^k \rangle = 0. \tag{69}$$
Rearranging the terms in the above equation, we have
$$\mathbf{c}^T\mathbf{x}^{k+1} + \mathbf{b}^T\mathbf{z}_\mathbf{x}^k = \langle \mathbf{A}_1\mathbf{x}^{k+1} - \mathbf{z}^k/\rho, \mathbf{p}^k - \mathbf{p}^{k+1} \rangle. \tag{70}$$
Thus, the lemma follows.

# D Proof of Lemma 5

Let $\mathcal{S}^*$ denote the solution set described by Lemma 2. Since $\mathcal{S}^*$ is a non-empty polyhedron, we can utilize the Hoffman bound in Lemma 3 to bound the distance between the primal, dual iterates and the optimal solution set $\mathcal{S}^*$.

$$
\left\| \begin{bmatrix} \mathbf{x}^{k+1} \\ \mathbf{y}^k \\ \mathbf{z}^k \end{bmatrix} - \left[ \begin{bmatrix} \mathbf{x}^{k+1} \\ \mathbf{y}^k \\ \mathbf{z}^k \end{bmatrix} \right]_{\mathcal{S}^*} \right\| \leq \theta_{\mathcal{S}^*} \left\| \begin{bmatrix} \mathbf{A}_1 \mathbf{x}^{k+1} + \mathbf{A}_2 \mathbf{y}^{k+1} - \overline{\mathbf{b}} \\ -\mathbf{A}_1^T \mathbf{z}^k - \mathbf{c} \\ [-\mathbf{y}^k]_+ \\ [\mathbf{z}_y^k]_+ \\ \mathbf{c}^T \mathbf{x}^{k+1} + \mathbf{b}^T \mathbf{z}_x^k \end{bmatrix} \right\|
$$

$$
\overset{(a)}{=} \theta_{\mathcal{S}^*} \left\| \begin{bmatrix} \mathbf{A}_1 \mathbf{x}^{k+1} + \mathbf{A}_2 \mathbf{y}^{k+1} - \overline{\mathbf{b}} \\ -\mathbf{A}_1^T \mathbf{z}^k - \mathbf{c} \\ \mathbf{c}^T \mathbf{x}^{k+1} + \mathbf{b}^T \mathbf{z}_x^k \end{bmatrix} \right\|
$$

$$
\overset{(b)}{=} \theta_{\mathcal{S}^*} \left\| \begin{bmatrix} (\mathbf{p}^{k+1} - \mathbf{p}^k)/\rho \\ \mathbf{A}_1^T (\mathbf{p}^{k+1} - \mathbf{p}^k) \\ \left\langle \mathbf{A}_1 \mathbf{x}^{k+1} - \mathbf{z}_x^k/\rho, \mathbf{p}^k - \mathbf{p}^{k+1} \right\rangle \end{bmatrix} \right\|
$$

$$
\overset{(c)}{\leq} \theta_{\mathcal{S}^*} \left( \|\mathbf{p}^{k+1} - \mathbf{p}^k\|/\rho + \|\mathbf{A}_1^T (\mathbf{p}^{k+1} - \mathbf{p}^k)\| \right) +
$$

$$
\theta_{\mathcal{S}^*} \left\| \left\langle \mathbf{A}_1 \mathbf{x}^{k+1} - \mathbf{z}_x^k/\rho, \mathbf{p}^k - \mathbf{p}^{k+1} \right\rangle \right\|
$$

$$
\overset{(d)}{\leq} \theta_{\mathcal{S}^*} \left[ (1 + R_z)/\rho + (R_x + 1)\|\mathbf{A}_1^T\| \right] \|\mathbf{p}^{k+1} - \mathbf{p}^k\|. \tag{71}
$$

The above, (a) is based on (64) and (65) such that $[-\mathbf{y}^k]_+ = 0$ and $[\mathbf{z}_y^k]_+ = 0$ (Note that the projection operator $[\cdot]_+$ is elementary wise and we omit the constraints that $z_{y,i} = 0, i \in [n] \backslash [n_b]$ since it is always satisfied (Lemma 4), (b) is based on the estimation of residuals in Lemma 4, (c) utilizes the triangle inequality, (d) utilizes the following spectrum inequality

$$
\|\mathbf{A}_1^T x\| \leq \|\mathbf{A}_1^T\| \|\mathbf{x}\|, \tag{72}
$$

where $\|\mathbf{A}_1^T\|$ is the spectral norm of matrix $\mathbf{A}_1^T$, defined as $\|\mathbf{A}_1^T\|^2 = \rho_{max}(\mathbf{A}_1 \mathbf{A}_1^T)$ (the maximum eigenvalue of matrix $\mathbf{A}_1 \mathbf{A}_1^T$). Besides,

$$
\left\| \left\langle \mathbf{A}_1 \mathbf{x}^{k+1} - \frac{1}{\rho} \mathbf{z}_x^k, \mathbf{p}^k - \mathbf{p}^{k+1} \right\rangle \right\| = \left\| \left\langle \mathbf{x}^{k+1}, \mathbf{A}_1^T (\mathbf{p}^k - \mathbf{p}^{k+1}) \right\rangle - \left\langle \mathbf{z}_x^k/\rho, \mathbf{p}^k - \mathbf{p}^{k+1} \right\rangle \right\|
$$

$$
\overset{(e)}{\leq} \left\| \left\langle \mathbf{x}^{k+1}, \mathbf{A}_1^T (\mathbf{p}^k - \mathbf{p}^{k+1}) \right\rangle \right\| + \left\| \left\langle \mathbf{z}_x^k/\rho, \mathbf{p}^k - \mathbf{p}^{k+1} \right\rangle \right\|
$$

$$
\overset{(f)}{\leq} R_x \|\mathbf{A}_1^T\| \|\mathbf{p}^k - \mathbf{p}^{k+1}\| + \frac{R_z}{\rho} \|\mathbf{p}^k - \mathbf{p}^{k+1}\|. \tag{73}
$$

The above, (e) is based on the triangle inequality, (f) utilizes Cauchy-Schwarz inequality and spectrum inequality. Here $R_x$ and $R_z$ is defined as

$$
R_x = \sup_k \|\mathbf{x}^k\| \quad \text{and} \quad R_z = \sup_k \|\mathbf{z}_x^k\|. \tag{74}
$$

Based on the above results, we have the following two inequalities,

$$
\|\mathbf{y}^k - [\mathbf{y}^k]_{\mathcal{S}^*}\| \leq \gamma' \|\mathbf{p}^{k+1} - \mathbf{p}^k\| \text{ and } \|\mathbf{z}^k - [\mathbf{z}^k]_{\mathcal{S}^*}\| \leq \gamma' \|\mathbf{p}^{k+1} - \mathbf{p}^k\|, \tag{75}
$$

where $[\mathbf{x}^{k+1}]_{\mathcal{S}^*}$, $[\mathbf{y}^k]_{\mathcal{S}^*}$ and $[\mathbf{z}^k]_{\mathcal{S}^*}$ are the sub-vector of the $\left[ \begin{bmatrix} \mathbf{x}^{k+1} \\ \mathbf{y}^k \\ \mathbf{z}^k \end{bmatrix} \right]_{\mathcal{S}^*}$ with corresponding coordinates of $\mathbf{x}$, $\mathbf{y}$ and $\mathbf{z}$, and the estimation $\gamma' = \left[ (R_z + 1)/\rho + (R_x + 1)\|\mathbf{A}_1^T\| \right] \theta_{\mathcal{S}^*}$.

**Claim**: $\mathbf{p}^* = [\mathbf{z}^k]_{\mathcal{S}^*} - \rho \mathbf{A}_2 [\mathbf{y}^k]_{\mathcal{S}^*}$ belongs to the optimal solution set $G^*$, that is $G(\mathbf{p}^*) = 0$ (defined in Lemma 1).

Since $([\mathbf{x}^{k+1}]_{\mathcal{S}^*}, [\mathbf{y}^k]_{\mathcal{S}^*}, [\mathbf{z}^k]_{\mathcal{S}^*})$ are the optimal primal and dual solutions of LPs (7) and (8), they satisfy the following conditions.

1. Primal feasibility: $\mathbf{A}_1[\mathbf{x}^{k+1}]_{\mathcal{S}^*} + \mathbf{A}_2[\mathbf{y}^k]_{\mathcal{S}^*} = \overline{\mathbf{b}}$;
2. Dual feasibility: $\mathbf{c} + A_1^T[\mathbf{z}^k]_{\mathcal{S}^*} = \mathbf{0}$;
3. Complementary slackness: $\langle -\mathbf{A}_2^T[\mathbf{z}^k]_{\mathcal{S}^*}, [\mathbf{y}^k]_{\mathcal{S}^*}\rangle = \mathbf{0}$.

Then, we have

$$
\begin{aligned}
G(\mathbf{p}^*) &= \mathbf{prox}_{\rho\overline{g}}(\mathbf{p}^*) - \mathbf{prox}_{\rho\overline{f}}(2\mathbf{prox}_{\rho\overline{g}}(\mathbf{p}^*) - \mathbf{p}^*) \\
&\overset{(a)}{=} [\mathbf{z}^k]_{\mathcal{S}^*} - \mathbf{prox}_{\rho\overline{f}}(2[\mathbf{z}^k]_{\mathcal{S}^*} - \mathbf{p}^*) \\
&\overset{(b)}{=} [\mathbf{z}^k]_{\mathcal{S}^*} - \mathbf{prox}_{\rho\overline{f}}([\mathbf{z}^k]_{\mathcal{S}^*} + \rho\mathbf{A}_2[\mathbf{y}^k]_{\mathcal{S}^*}) \\
&\overset{(c)}{=} \mathbf{0}.
\end{aligned}
$$

The above, (a) is based on the following argument

$$
\begin{aligned}
[\mathbf{z}^k]_{\mathcal{S}^*} = \mathbf{prox}_{\rho\overline{g}}(\mathbf{p}^*) &\overset{(d)}{\Leftarrow} \mathbf{0} \in -\mathbf{A}_2\partial g^*(-\mathbf{A}_2^T[\mathbf{z}^k]_{\mathcal{S}^*}) + ([\mathbf{z}^k]_{\mathcal{S}^*} - \mathbf{p}^*)/\rho \\
&\overset{(e)}{\Leftarrow} \mathbf{A}_2[\mathbf{y}^k]_{\mathcal{S}^*} \in \mathbf{A}_2\partial g^*(-\mathbf{A}_2^T[\mathbf{z}^k]_{\mathcal{S}^*}) \\
&\overset{(f)}{\Leftarrow} -\mathbf{A}_2^T[\mathbf{z}^k]_{\mathcal{S}^*} \in \partial g([\mathbf{y}^k]_{\mathcal{S}^*}) \\
&\overset{(g)}{\Leftarrow} \langle -\mathbf{A}_2^T[\mathbf{z}^k]_{\mathcal{S}^*}, [\mathbf{y}^k]_{\mathcal{S}^*}\rangle \geq \langle -\mathbf{A}_2^T[\mathbf{z}^k]_{\mathcal{S}^*}, \mathbf{y}\rangle, \forall y_i \geq 0, i \in [n_b].
\end{aligned}
$$

The above, (d) is based on the first-order optimality condition of the proximal operator, (e) utilizes the definition of $\mathbf{p}^*$, (f) is based on full column rank property of matrix $\mathbf{A}_2$, (g) is based on the definition of the subgradients of the indicator function $g(\mathbf{y})$. The last inequality always holds because the left-hand-side is equal to 0 by complementary slackness (condition 3), and the right-hand-side is negative by definition of $\mathcal{S}^*$.

The step (b) utilizes the definition of $\mathbf{p}^*$. The step (c) is based on the following argument

$$
\begin{aligned}
[\mathbf{z}^k]_{\mathcal{S}^*} = \mathbf{prox}_{\rho\overline{f}}([\mathbf{z}^k]_{\mathcal{S}^*} + \rho\mathbf{A}_2[\mathbf{y}^k]_{\mathcal{S}^*}) &\overset{(h)}{\Leftarrow} \mathbf{0} \in \overline{\mathbf{b}} - \mathbf{A}_1\partial f^*(-\mathbf{A}_1^T[\mathbf{z}^k]_{\mathcal{S}^*}) + \\
&\qquad ([\mathbf{z}^k]_{\mathcal{S}^*} - [\mathbf{z}^k]_{\mathcal{S}^*} - \rho\mathbf{A}_2[\mathbf{y}^k]_{\mathcal{S}^*})/\rho \\
&\overset{(i)}{\Leftarrow} -\mathbf{A}_1[\mathbf{x}^{k+1}]_{\mathcal{S}^*} \in -\mathbf{A}_1\partial f^*(-\mathbf{A}_1^T[\mathbf{z}^k]_{\mathcal{S}^*}) \\
&\overset{(j)}{\Leftarrow} -\mathbf{A}_1^T[\mathbf{z}^k]_{\mathcal{S}^*} \in \partial f([\mathbf{x}^{k+1}]_{\mathcal{S}^*}).
\end{aligned}
$$

The above, (h) is based on the first-order optimality condition of the proximal operator, (i) utilizes the primal feasibility condition $\mathbf{A}_1[\mathbf{x}^{k+1}]_{\mathcal{S}^*} + \mathbf{A}_2[\mathbf{y}^k]_{\mathcal{S}^*} = \overline{\mathbf{b}}$, (j) is based on the similar argument of (f). The last equality always holds by dual feasibility condition $\mathbf{c} + \mathbf{A}_1^T[\mathbf{z}^k]_{\mathcal{S}^*} = \mathbf{0}$ and fact that $\partial f(\mathbf{x}) = \{\mathbf{c}\}$. Thus the claim follows.

Then, we have

$$
\begin{aligned}
\|\mathbf{p}^k - [\mathbf{p}^k]_{G^*}\| &\overset{(a)}{\leq} \|\mathbf{p}^k - \mathbf{p}^*\| \\
&= \|\mathbf{z}^k - [\mathbf{z}^k]_{\mathcal{S}^*} - \rho\mathbf{A}_2(\mathbf{y}^k - [\mathbf{y}^k]_{\mathcal{S}^*})\| \\
&\overset{(b)}{\leq} \|\mathbf{z}^k - [\mathbf{z}^k]_{\mathcal{S}^*}\| + \rho\|\mathbf{y}^k - [\mathbf{y}^k]_{\mathcal{S}^*}\| \\
&\overset{(c)}{\leq} (1+\rho)\gamma'\|\mathbf{p}^k - \mathbf{p}^{k+1}\|.
\end{aligned}
\tag{76}
$$

The above, (a) is based on definition of projection operator $[\cdot]_{G^*}$ and the claim that $\mathbf{p}^*$ belongs to $G^*$, (b) utilizes the triangle inequality and the definition of matrix $\mathbf{A}_2$, (c) utilizes results of (75). Therefore, the lemma follows.

# E    Proof of Theorem 1

We first prove the following convergence result each subproblem is exactly solved ($\epsilon_k = 0$).

**Lemma 7.** (Linear convergence of Algorithm 1 with exact subproblem solver) *Denote $\mathbf{z}^k$ as the dual iterates produced by Algorithm 1. In each iteration $k$, if the accuracy $\epsilon_k = 0$ and $k \geq 2\gamma^2 \log(D_0/\epsilon)$, then there exists an optimal dual solution $\mathbf{z}^*$ such that $\|\mathbf{z}^{k+1} - \mathbf{z}^*\| \leq \epsilon$, where $D_0 = \|\mathbf{p}^0 - [\mathbf{p}^0]_{G^*}\|$.*

*Proof.* We first show the accuracy of $\|\mathbf{z}^k - \mathbf{z}^*\|$. Combining the results of Lemma 1 and Lemma 5, we have

$$\|\mathbf{p}^{k+1} - [\mathbf{p}^{k+1}]_{G^*}\|^2 \leq \left(1 - \frac{1}{\gamma^2}\right)\|\mathbf{p}^k - [\mathbf{p}^k]_{G^*}\|^2. \tag{77}$$

Further,

$$\|\mathbf{p}^{k+1} - [\mathbf{p}^{k+1}]_{G^*}\| \leq \sqrt{1 - \frac{1}{\gamma^2}} \cdot \|\mathbf{p}^k - [\mathbf{p}^k]_{G^*}\|. \tag{78}$$

Then, telescoping (78), we have

$$\|\mathbf{p}^k - [\mathbf{p}^k]_{G^*}\| \leq \left(1 - \frac{1}{\gamma^2}\right)^{\frac{k}{2}} \cdot \|\mathbf{p}^0 - [\mathbf{p}^0]_{G^*}\|. \tag{79}$$

Thus, let the number of iterations $k$ satisfies

$$k \geq 2\gamma^2 \log\left(\frac{D_0}{\epsilon}\right), \tag{80}$$

where the constant $D_0$ is the distance between the initial point and optimal solution set, defined as $D_0 = \|\mathbf{p}^0 - [\mathbf{p}^0]_{G^*}\|$. Then we have

$$\begin{aligned}
\|\mathbf{p}^k - [\mathbf{p}^k]_{G^*}\| &\leq \left(1 - \frac{1}{\gamma^2}\right)^{\gamma^2 \log\left(\frac{D_0}{\epsilon}\right)} \cdot \|\mathbf{p}^0 - [\mathbf{p}^0]_{G^*}\| \\
&= \exp\left\{\gamma^2 \log\left(\frac{\epsilon}{D_0}\right)\log\left(\frac{\gamma^2}{\gamma^2 - 1}\right) + \log(D_0)\right\} \\
&\overset{(a)}{\leq} \exp\left\{\log\left(\frac{\epsilon}{D_0}\right) + \log(D_0)\right\} = \epsilon,
\end{aligned} \tag{81}$$

The above, (a) is based on the inequality: $\gamma^2 \log\left(\frac{\gamma^2}{\gamma^2-1}\right) \geq 1$, when $\gamma > 1$. Here the fact that $\gamma > 1$ derives from the result of Lemma 1.

We then show that the distance between the dual iterates $\mathbf{z}^k$ and the optimal solution set is also bounded by $\epsilon$, when $k$ satisfies the condition (80). Based on the first-order optimality condition, we have

$$\begin{aligned}
\mathbf{z}^k = \mathbf{prox}_{\rho\bar{g}}(\mathbf{p}^k) &\overset{(a)}{\Longleftrightarrow} -\mathbf{A}_2\mathbf{y}^k \in -\mathbf{A}_2 \partial g^*(-\mathbf{A}_2\mathbf{z}^k) \\
&\overset{(b)}{\Longleftrightarrow} \mathbf{y}^k \in \partial g^*(-\mathbf{A}_2\mathbf{z}^k).
\end{aligned} \tag{82}$$

The above, (a) is based on the definition of $\mathbf{p}^k = \mathbf{z}^k - \rho\mathbf{A}_2\mathbf{y}^k$, (b) utilizes the fact that matrix $\mathbf{A}_2$ has full column rank. The last equality is indeed the second step of the ADMM, as indicated in (45). Based on the definition of optimal solution set $G^*$, there exists an optimal primal and dual solution of LP $\mathbf{y}^*$ and $\mathbf{z}^*$ such that $[\mathbf{p}^k]_{G^*} = \mathbf{z}^* - \rho\mathbf{A}_2\mathbf{y}^*$. Similarly, we have that $\mathbf{z}^* = \mathbf{prox}_{\rho\bar{g}}([\mathbf{p}^k]_{G^*})$. According to the non-expansiveness of the proximal operator, we have there exists optimal multiplier $\mathbf{z}^*$ of ADMM such that

$$\|\mathbf{z}^k - \mathbf{z}^*\| = \|\mathbf{prox}_{\rho\bar{g}}(\mathbf{p}^k) - \mathbf{prox}_{\rho\bar{g}}([\mathbf{p}^k]_{G^*})\| \leq \|\mathbf{p}^k - [\mathbf{p}^k]_{G^*}\|. \tag{83}$$

Hence, if $k \geq 2\gamma^2 \log\left(\frac{D_0}{\epsilon}\right)$, then $\|\mathbf{z}^k - \mathbf{z}^*\| \leq \epsilon$.

Secondly, we show the accuracy of $\|\mathbf{x}^k - \mathbf{x}^*\|$ and $\|\mathbf{y}^k - \mathbf{y}^*\|$. From the result of (75), we have

$$\|\mathbf{x}^{k+1} - [\mathbf{x}^{k+1}]_{S^*}\| \leq \gamma'\|\mathbf{p}^{k+1} - \mathbf{p}^k\| \quad \text{and} \quad \|\mathbf{y}^k - [\mathbf{y}^k]_{S^*}\| \leq \gamma'\|\mathbf{p}^{k+1} - \mathbf{p}^k\|. \tag{84}$$

According to Lemma 1, we have

$$\|\mathbf{p}^{k+1} - \mathbf{p}^k\| \leq \|\mathbf{p}^k - [\mathbf{p}^k]_{G^*}\|.$$

Thus, to guarantee that both $\|\mathbf{x}^{k+1} - [\mathbf{x}^{k+1}]_{\mathcal{S}^*}\| \leq \epsilon$ and $\|\mathbf{y}^k - [\mathbf{y}^k]_{\mathcal{S}^*}\| \leq \epsilon$, we requires

$$k \geq 2\gamma^2 \log\left(\frac{D_0 \gamma}{(1+\rho)\epsilon}\right).$$

The accuracy of the duality gap is related to the accuracy of both the primal and dual iterates by

$$\|\mathbf{c}^T\mathbf{x}^{k+1} + \mathbf{b}^T z_x^k\| \overset{(a)}{=} \|\mathbf{c}^T\mathbf{x}^{k+1} - \mathbf{c}^T\mathbf{x}^* - \mathbf{b}^T\mathbf{z}_x^* + \mathbf{b}^T z_x^k\| \leq \|\mathbf{c}^T\mathbf{x}^{k+1} - \mathbf{c}^T\mathbf{x}^*\| + \|\mathbf{b}^T\mathbf{z}^* - \mathbf{b}^T z_x^k\|$$

$$\overset{(c)}{\leq} \|\mathbf{c}\| \cdot \|\mathbf{x}^{k+1} - \mathbf{x}^*\| + \|\mathbf{b}\| \cdot \|\mathbf{z}_x^k - z_{\mathbf{x}}^*\|.$$

The above, (a) is based on the strong duality theorem that $\mathbf{c}^T\mathbf{x}^* + \mathbf{b}^T\mathbf{z}_x^* = 0$, (c) follows from the Cauchy-Schwarz inequality. Thus, the lemma follows. $\qquad\square$

We next generalize the convergence result to the inexact subproblem solver ($\epsilon_k > 0$). Let the iterates under the inexact update be denoted by $\overline{\mathbf{x}}^k, \overline{\mathbf{y}}^k, \overline{\mathbf{z}}^k, \overline{\mathbf{p}}^k = \overline{\mathbf{z}}^k - \rho\mathbf{A}_2\overline{\mathbf{y}}^k$ and corresponding AL function

$$\overline{F}_k(\mathbf{x}) = \mathbf{c}^T\mathbf{x} + (\overline{\mathbf{z}}^k)^T\mathbf{A}_1\mathbf{x} + \frac{\rho}{2}\|\mathbf{A}_1\mathbf{x} + \mathbf{A}_2\overline{\mathbf{y}}^k - \overline{\mathbf{b}}\|^2. \tag{85}$$

We first construct the relation between the primal accuracy $\epsilon_k$ and the accuracy of the dual iterates in the subproblem (11) by a standard primal dual argument; then we connect the accuracy of such dual iterates with the $\overline{\mathbf{p}}^k$.

**Lemma 8.** (Relation between primal and dual accuracy) *Let* $\overline{\mathbf{w}}^{k+1} = \overline{\mathbf{z}}^k + \rho(\mathbf{A}_1\overline{\mathbf{x}}^{k+1} + \mathbf{A}_2\overline{\mathbf{y}}^k - \overline{\mathbf{b}})$. *If the* $\overline{F}_k(\overline{\mathbf{x}}^{k+1}) - \min_x \overline{F}_k(\mathbf{x}) \leq \epsilon_k$, *then the dual iterates satisfy.*

$$\|\overline{\mathbf{w}}^{k+1} - \mathbf{prox}_{\rho\overline{f}}(\overline{\mathbf{z}}^k + \rho\mathbf{A}_2\overline{\mathbf{y}}^k)\| \leq \sqrt{2\rho\epsilon_k}.$$

*Proof.* The quadratic function with inexact update in the step 1 of Algorithm 1 is

$$\overline{F}_k(\mathbf{x}) \triangleq f(\mathbf{x}) + (\overline{\mathbf{z}}^k)^T(\mathbf{A}_1\mathbf{x} - \overline{\mathbf{b}}) + \frac{\rho}{2}\|\mathbf{A}_1\mathbf{x} + \mathbf{A}_2\overline{\mathbf{y}}^k - \overline{\mathbf{b}}\|^2. \tag{86}$$

The proof of this lemma is based on the following two claims.

**Claim 1**: The following two problems are primal and dual optimization problems.

$$\text{primal:} \min_{\mathbf{x}} \overline{F}_k(\mathbf{x}) - \frac{\rho}{2}\|\mathbf{A}_2\overline{\mathbf{y}}^k\|^2 \iff \text{dual:} \max_{\mathbf{w}} -\overline{\mathbf{b}}^T\mathbf{w} - f^*(-\mathbf{A}_1^T\mathbf{w}) - \frac{1}{2\rho}\|\mathbf{w} - \overline{\mathbf{z}}^k - \rho\mathbf{A}_2\overline{\mathbf{y}}^k\|^2, \tag{87}$$

Let $\mathbf{A}_1\mathbf{x} + \mathbf{A}_2\overline{\mathbf{y}}^k - \overline{\mathbf{b}} = \overline{\mathbf{t}}$, then the primal problem is equivalent to

$$\min_{x, \overline{\mathbf{t}}} \quad f(\mathbf{x}) + (\overline{\mathbf{z}}^k)^T(\overline{\mathbf{t}} - \mathbf{A}_2\overline{\mathbf{y}}^k) + \frac{\rho}{2}\|\overline{\mathbf{t}}\|^2 - \frac{\rho}{2}\|\mathbf{A}_2\overline{\mathbf{y}}^k\|^2 \tag{88}$$

$$s.t. \quad \mathbf{A}_1\mathbf{x} + \mathbf{A}_2\overline{\mathbf{y}}^k - \overline{\mathbf{b}} = \overline{\mathbf{t}}.$$

Then, the dual optimization problem can be written as minimizing the Lagrangian function w.r.t $x$ and $\overline{\mathbf{t}}$.

$$\min_{x, \overline{\mathbf{t}}} f(\mathbf{x}) + (\overline{\mathbf{z}}^k)^T(\overline{\mathbf{t}} - \mathbf{A}_2\overline{\mathbf{y}}^k) + \frac{\rho}{2}\|\overline{\mathbf{t}}\|^2 + w^T(\mathbf{A}_1\mathbf{x} + \mathbf{A}_2\overline{\mathbf{y}}^k - \overline{\mathbf{b}} - \overline{\mathbf{t}}) - \frac{\rho}{2}\|\mathbf{A}_2\overline{\mathbf{y}}^k\|^2$$

$$\overset{(a)}{=} \min_x \left[f(\mathbf{x}) + \langle\mathbf{A}_1^T w, \mathbf{x}\rangle\right] + \min_{\overline{\mathbf{t}}} \left[(\overline{\mathbf{z}}^k)^T(\overline{\mathbf{t}} - \mathbf{A}_2\overline{\mathbf{y}}^k) + \frac{\rho}{2}\|\overline{\mathbf{t}}\|^2 + w^T(\mathbf{A}_2\overline{\mathbf{y}}^k - \overline{\mathbf{b}} - \overline{\mathbf{t}}) - \frac{\rho}{2}\|\mathbf{A}_2\overline{\mathbf{y}}^k\|^2\right]$$

$$\overset{(b)}{=} -\overline{\mathbf{b}}^T w - f^*(-\mathbf{A}_1^T w) + \min_{\overline{\mathbf{t}}} \left[(\overline{\mathbf{z}}^k)^T(\overline{\mathbf{t}} - \mathbf{A}_2\overline{\mathbf{y}}^k) + \frac{\rho}{2}\|\overline{\mathbf{t}}\|^2 + w^T(\mathbf{A}_2\overline{\mathbf{y}}^k - \overline{\mathbf{t}}) - \frac{\rho}{2}\|\mathbf{A}_2\overline{\mathbf{y}}^k\|^2\right]$$

$$\overset{(c)}{=} -\overline{\mathbf{b}}^T w - f^*(-\mathbf{A}_1^T w) - \frac{1}{2\rho}\|w - \overline{\mathbf{z}}^k - \rho\mathbf{A}_2\overline{\mathbf{y}}^k\|^2. \tag{89}$$

The above, (a) is based on the separability between variable $x$ and $\overline{\mathbf{t}}$ in the above optimization problem, (b) utilizes the definition of the convex conjugate function, (c) is obtained by setting $\overline{\mathbf{t}} = (w - \overline{\mathbf{z}}^k)/\rho$.

**Claim 2**: The following two problems are primal and dual optimization problems.

$$\text{primal:} \min_{\mathbf{x}} \tilde{F}_k(\mathbf{x}) \triangleq f(\mathbf{x}) + \mathbf{s}^T(\mathbf{A}_1\mathbf{x} - \overline{\mathbf{b}}) + \frac{\rho}{2}\|\mathbf{A}_1\mathbf{x} + \mathbf{A}_2\overline{\mathbf{y}}^k - \overline{\mathbf{b}}\|^2 - \frac{\rho}{2}\|\mathbf{A}_2\overline{\mathbf{y}}^k\|^2$$

$$\Longleftrightarrow \text{dual:} \max_{w} -\overline{\mathbf{b}}^T\mathbf{w} - f^*(-\mathbf{A}_1^T\mathbf{w}) - \frac{1}{2\rho}\|\mathbf{w} - \mathbf{s} - \rho\mathbf{A}_2\overline{\mathbf{y}}^k\|^2 \tag{90}$$

Let $\mathbf{A}_1\mathbf{x} + \mathbf{A}_2\overline{\mathbf{y}}^k - \overline{\mathbf{b}} = \overline{\mathbf{t}}$, then the primal problem is equivalent to

$$\min_{x,\overline{\mathbf{t}}} \quad f(\mathbf{x}) + s^T(\overline{\mathbf{t}} - \mathbf{A}_2\overline{\mathbf{y}}^k) + \frac{\rho}{2}\|\overline{\mathbf{t}}\|^2 - \frac{\rho}{2}\|\mathbf{A}_2\overline{\mathbf{y}}^k\|^2 \tag{91}$$

$$\text{s.t.} \quad \mathbf{A}_1\mathbf{x} + \mathbf{A}_2\overline{\mathbf{y}}^k - \overline{\mathbf{b}} = \overline{\mathbf{t}}.$$

Then, the dual optimization problem can be written as minimizing the Lagrangian function w.r.t $x$ and $\overline{\mathbf{t}}$.

$$\min_{x,\overline{\mathbf{t}}} f(\mathbf{x}) + \mathbf{s}^T(\overline{\mathbf{t}} - \mathbf{A}_2\overline{\mathbf{y}}^k) + \frac{\rho}{2}\|\overline{\mathbf{t}}\|^2 + \mathbf{w}^T(\mathbf{A}_1\mathbf{x} + \mathbf{A}_2\overline{\mathbf{y}}^k - \overline{\mathbf{b}} - \overline{\mathbf{t}}) - \frac{\rho}{2}\|\mathbf{A}_2\overline{\mathbf{y}}^k\|^2$$

$$\overset{(a)}{=} \min_{x}\left[f(\mathbf{x}) + \langle\mathbf{A}_1^T\mathbf{w},\mathbf{x}\rangle\right] + \min_{\overline{\mathbf{t}}}\left[\mathbf{s}^T(\overline{\mathbf{t}} - \mathbf{A}_2\overline{\mathbf{y}}^k) + \frac{\rho}{2}\|\overline{\mathbf{t}}\|^2 + \mathbf{w}^T(\mathbf{A}_2\overline{\mathbf{y}}^k - \overline{\mathbf{b}} - \overline{\mathbf{t}})\right] - \frac{\rho}{2}\|\mathbf{A}_2\overline{\mathbf{y}}^k\|^2$$

$$\overset{(b)}{=} -\overline{\mathbf{b}}^T\mathbf{w} - f^*(-\mathbf{A}_1^T\mathbf{w}) + \min_{\overline{\mathbf{t}}}\left[\mathbf{s}^T(\overline{\mathbf{t}} - \mathbf{A}_2\overline{\mathbf{y}}^k) + \frac{\rho}{2}\|\overline{\mathbf{t}}\|^2 + \mathbf{w}^T(\mathbf{A}_2\overline{\mathbf{y}}^k - \overline{\mathbf{t}})\right] - \frac{\rho}{2}\|\mathbf{A}_2\overline{\mathbf{y}}^k\|^2$$

$$\overset{(c)}{=} -\overline{\mathbf{b}}^T\mathbf{w} - f^*(-\mathbf{A}_1^T\mathbf{w}) - \frac{1}{2\rho}\|\mathbf{w} - \mathbf{s} - \rho\mathbf{A}_2\overline{\mathbf{y}}^k\|^2. \tag{92}$$

The above, (a) is based on the separability between variable $x$ and $\overline{\mathbf{t}}$ in the above optimization problem, (b) utilizes the definition of the convex conjugate function, (c) utilizes the first-order optimality condition of the second optimization problem such that $\overline{\mathbf{t}} = (\mathbf{w} - \mathbf{s})/\rho$. Define the following iterates,

$$\tilde{\mathbf{x}}^{k+1} = \arg\min_{x} \overline{F}_k(\mathbf{x}) - \frac{\rho}{2}\|\mathbf{A}_2\mathbf{y}^k\|^2 \text{ and } \tilde{\mathbf{w}}^{k+1} = \mathbf{prox}_{\rho\overline{f}}(\overline{\mathbf{z}}^k + \rho\mathbf{A}_2\overline{\mathbf{y}}^k). \tag{93}$$

Then the pair of sequences $\tilde{\mathbf{x}}^{k+1}$ and $\tilde{w}^{k+1}$ are the optimal primal and dual solutions of (87). According to the strong duality theorem of the convex optimization, we have

$$\overline{F}_k(\tilde{\mathbf{x}}^{k+1}) - \frac{\rho}{2}\|\mathbf{A}_2\overline{\mathbf{y}}^k\|^2 = -\overline{\mathbf{b}}^T\tilde{\mathbf{w}}^{k+1} - f^*(-\mathbf{A}_1^T\tilde{\mathbf{w}}^{k+1}) - \frac{1}{2\rho}\|\tilde{\mathbf{w}}^{k+1} - \overline{\mathbf{z}}^k - \rho\mathbf{A}_2\overline{\mathbf{y}}^k\|^2. \tag{94}$$

The pair of sequences $\overline{\mathbf{x}}^{k+1}$ and $\tilde{\mathbf{w}}^{k+1}$ is the primal and dual feasible solution of problem (90). According to the weak duality theorem of the convex optimization, we have

$$\tilde{F}_k(\overline{\mathbf{x}}^{k+1}) \geq -\overline{\mathbf{b}}^T\tilde{\mathbf{w}}^{k+1} - f^*(-\mathbf{A}_1^T\tilde{\mathbf{w}}^{k+1}) - \frac{1}{2\rho}\|\tilde{\mathbf{w}}^{k+1} - \mathbf{s} - \rho\mathbf{A}_2\overline{\mathbf{y}}^k\|^2$$

Further, based on the definition of $\overline{F}_k(\mathbf{x})$ and $\tilde{F}^k(\mathbf{x})$, we have

$$\overline{F}_k(\overline{\mathbf{x}}^{k+1}) - \frac{\rho}{2}\|\mathbf{A}_2\overline{\mathbf{y}}^k\|^2 \geq (\overline{\mathbf{z}}^k - \mathbf{s})^T(\mathbf{A}_1\overline{\mathbf{x}}^{k+1} - \overline{\mathbf{b}}) - \overline{\mathbf{b}}^T\tilde{\mathbf{w}}^{k+1} - f^*(-\mathbf{A}_1^T\tilde{\mathbf{w}}^{k+1}) -$$

$$\frac{1}{2\rho}\|\tilde{\mathbf{w}}^{k+1} - \mathbf{s} - \rho\mathbf{A}_2\overline{\mathbf{y}}^k\|^2 \tag{95}$$

Combining the results of (94) and (95), we have

$$\overline{F}_k(\overline{\mathbf{x}}^{k+1}) - \overline{F}_k(\tilde{\mathbf{x}}^{k+1}) \geq \frac{1}{2\rho}\|\tilde{\mathbf{w}}^{k+1} - \overline{\mathbf{z}}^k - \rho\mathbf{A}_2\overline{\mathbf{y}}^k\|^2 - \frac{1}{2\rho}\|\tilde{\mathbf{w}}^{k+1} - \mathbf{s} - \rho\mathbf{A}_2\overline{\mathbf{y}}^k\|^2 +$$

$$(\overline{\mathbf{z}}^k - \mathbf{s})^T(\mathbf{A}_1\overline{\mathbf{x}}^{k+1} - \overline{\mathbf{b}})$$

$$\overset{(a)}{\geq} \frac{1}{2\rho}\|\overline{\mathbf{w}}^{k+1} - \mathbf{prox}_{\rho\overline{f}}(\overline{\mathbf{z}}^k + \rho\mathbf{A}_2\overline{\mathbf{y}}^k)\|^2. \tag{96}$$

The above, (a) utilizes the definition of $\overline{\mathbf{w}}^{k+1} = \overline{\mathbf{z}}^k + \rho(\mathbf{A}_1\overline{\mathbf{x}}^{k+1} + \mathbf{A}_2\overline{\mathbf{y}}^k - \overline{\mathbf{b}})$ to substitute $\mathbf{A}_1\overline{\mathbf{x}}^{k+1} - \overline{\mathbf{b}}$, and maximizing the righthand side w.r.t $s$. Thus, the lemma follows. $\quad\square$

Utilizing the requirement that $\overline{F}_k(\overline{\mathbf{x}}^{k+1}) - \min_{\mathbf{x}} \overline{F}_k(\mathbf{x}) \le \epsilon_k$ in the Algorithm 1 and the result in Lemma 8, we have the following inexact version of proximal operations.

$$\|\overline{\mathbf{w}}^{k+1} - \mathbf{prox}_{\rho\overline{f}}(2\overline{\mathbf{z}}^k - \overline{\mathbf{p}}^k)\| \le \sqrt{2\rho\epsilon_k}, \tag{97}$$

$$\overline{\mathbf{z}}^{k+1} = \mathbf{prox}_{\rho\overline{g}}(\overline{\mathbf{w}}^{k+1} - \overline{\mathbf{z}}^k + \overline{\mathbf{p}}^k), \tag{98}$$

$$\overline{\mathbf{p}}^{k+1} = \overline{\mathbf{p}}^k + \overline{\mathbf{w}}^{k+1} - \overline{\mathbf{z}}^k. \tag{99}$$

The (97)-(99) can be simplified as

$$\|\overline{\mathbf{p}}^{k+1} - \overline{\mathbf{p}}^k + \overline{\mathbf{z}}^k - \mathbf{prox}_{\rho\overline{f}}(2\overline{\mathbf{z}}^k - \overline{\mathbf{p}}^k)\| \le \sqrt{2\rho\epsilon_k}, \tag{100}$$

$$\overline{\mathbf{z}}^{k+1} = \mathbf{prox}_{\rho\overline{g}}(\overline{\mathbf{p}}^{k+1}). \tag{101}$$

Start at $\mathbf{p}^0$ and renumber the iterates of (100) and (101), we have

$$\overline{\mathbf{z}}^{k+1} = \mathbf{prox}_{\rho\overline{g}}(\overline{\mathbf{p}}^k),$$

$$\|\overline{\mathbf{p}}^{k+1} - \overline{\mathbf{p}}^k + \overline{\mathbf{z}}^{k+1} - \mathbf{prox}_{\rho\overline{f}}(2\overline{\mathbf{z}}^{k+1} - \overline{\mathbf{p}}^k)\| \le \sqrt{2\rho\epsilon_k},$$

which can be further simplified as

$$\|\overline{\mathbf{p}}^{k+1} - [\overline{\mathbf{p}}^k - \mathbf{prox}_{\rho\overline{g}}(\overline{\mathbf{p}}^k) + \mathbf{prox}_{\rho\overline{f}}(2\mathbf{prox}_{\rho\overline{g}}(\overline{\mathbf{p}}^k) - \overline{\mathbf{p}}^k)]\| \le \sqrt{2\rho\epsilon_k}$$

$$\iff \|\overline{\mathbf{p}}^{k+1} - T(\overline{\mathbf{p}}^k)\| \le \sqrt{2\rho\epsilon_k}. \tag{102}$$

Then, we have

$$\begin{aligned} \|\overline{\mathbf{p}}^{k+1} - \mathbf{p}^{k+1}\| &\le \|\overline{\mathbf{p}}^{k+1} - T(\overline{\mathbf{p}}^k)\| + \|T(\overline{\mathbf{p}}^k) - \mathbf{p}^{k+1}\| \\ &\overset{(a)}{=} \|\overline{\mathbf{p}}^{k+1} - T(\overline{\mathbf{p}}^k)\| + \|T(\overline{\mathbf{p}}^k) - T(\mathbf{p}^k)\| \\ &\overset{(b)}{\le} \|\overline{\mathbf{p}}^{k+1} - T(\overline{\mathbf{p}}^k)\| + \|\overline{\mathbf{p}}^k - \mathbf{p}^k\| \\ &\overset{(c)}{\le} \sqrt{2\rho\epsilon_k} + \|\overline{\mathbf{p}}^k - \mathbf{p}^k\| \\ &\overset{(d)}{\le} \sum_{i=0}^{k} \sqrt{2\rho\epsilon_i}. \end{aligned} \tag{103}$$

The above, (a) is based on the definition that $T(\mathbf{p}^k) = \mathbf{p}^{k+1}$, (b) utilizes the non-expansiveness of the operator $T$, which can be derived from the Claim 4 in the proof of Lemma 1, (c) is based on the result in (102), (d) derives from telescoping the above inequality. Then, we can obtain

$$\|\overline{\mathbf{p}}^k - [\overline{\mathbf{p}}^k]_{G^*}\| \le \|\overline{\mathbf{p}}^k - [\mathbf{p}^k]_{G^*}\| \le \|\overline{\mathbf{p}}^k - \mathbf{p}^k\| + \|\mathbf{p}^k - [\mathbf{p}^k]_{G^*}\| \le \sum_{i=0}^{k-1} \sqrt{2\rho\epsilon_i} + \|\mathbf{p}^k - [\mathbf{p}^k]_{G^*}\|. \tag{104}$$

Let the solving accuracy $\epsilon_k$ of each iteration $k$ satisfies

$$\epsilon_k = \frac{\epsilon^2}{8\rho K^2}, K = 2\gamma^2 \log\left(\frac{2D_0}{\epsilon}\right), \forall k. \tag{105}$$

Then, we have

$$\|\overline{\mathbf{p}}^k - [\overline{\mathbf{p}}^k]_{G^*}\| \le \frac{\epsilon}{2} + \|\mathbf{p}^k - [\mathbf{p}^k]_{G^*}\|. \tag{106}$$

Combining the result (81) in Lemma 7, we finally arrive

$$\|\overline{\mathbf{p}}^k - [\overline{\mathbf{p}}^k]_{G^*}\| \le \frac{\epsilon}{2} + \frac{\epsilon}{2} = \epsilon, \text{ if } k \ge 2\gamma^2 \log\left(\frac{2D_0}{\epsilon}\right). \tag{107}$$

Utilizing a similar argument in the proof of Lemma 7, we can obtain the accuracy of both $\mathbf{x}^*, \mathbf{y}^*, \mathbf{z}^*$ and the duality gap. Therefore, the Theorem 1 follows.

# F Proof of Lemma 6

Based on the analysis in [22], to obtain an $\epsilon$ accurate solution, it requires running ACDM by $O(1/\tau \log(1/\epsilon))$ iterations, where

$$\tau = O\left(\frac{\sqrt{\rho}}{\sum\limits_{i=1}^{n} \sqrt{L_i}}\right). \tag{108}$$

According to the form of subproblem (11), the component-wise Lipschitz constant $L_i$ is equal to

$$L_i = \|\mathbf{A}_{1,i}\|^2 = \|\mathbf{A}_i\|^2 + 1. \tag{109}$$

Further,

$$\sum_{i=1}^{n} \sqrt{L_i} = \sum_{i=1}^{n} \sqrt{\|\mathbf{A}_i\|^2 + 1} = O(\|\mathbf{A}\|_{2,1}), \tag{110}$$

where $\|\mathbf{A}\|_{p,q} = (\sum_{j=1}^{n}(\sum_{i=1}^{m} |A_{ij}|^p)^{q/p})^{1/q}$ is the $L_{p,q}$ norm of constraint matrix $\mathbf{A}$. The $\rho$ is defined as the strongly convexity parameter of the function $\overline{F}_k(\mathbf{x})$. Based on the form of Hessian of function $\overline{F}_k(\mathbf{x})$ in (13), we have

$$\rho = \lambda_{min}(\rho(\mathbf{A}^T\mathbf{A} + \mathbf{I})) \geq \lambda_{min}(\rho\mathbf{A}^T\mathbf{A}) + \lambda_{min}(\rho\mathbf{I}) \geq \rho > 0. \tag{111}$$

Therefore, the iteration complexity is obtained by choosing parameter $\beta = 0$ in [22] and utilizing the Theorem 1 in [24] to transform the convergence in expectation to the form of probability.

In each iteration of Algorithm 2, the calculation of coordinate-wise gradient $\nabla_i F_k(\mathbf{u}_t)$ requires a vector product between $i$th column of matrix $\mathbf{A}$ and $\overline{\mathbf{u}}_t$, and the update of auxillary variables $\overline{\mathbf{u}}, \overline{\mathbf{v}}$ requires subtracting $i$th column of matrix $\mathbf{A}$. These two steps can be calculated in $O(nnz(\mathbf{A}_i))$ time. Therefore, the complexity of each step of ACDM is $O(nnz(\mathbf{A}_i))$.

Note that

$$\det(\mathbf{M}_t) = [\det(\mathbf{M})]^t = \left(\frac{1-\tau}{1+\eta\rho}\right)^t \tag{112}$$

In analysis in [22], we have

$$\eta = O\left(\frac{1}{\sqrt{\rho} \sum\limits_{i=1}^{n} \sqrt{L_i}}\right), \tag{113}$$

and the total number of iterations of ACDM is $O(1/\tau \log(1/\epsilon))$. Thus we have,

$$\det(\mathbf{M}_t) = \left(\frac{1-\tau}{1+\eta\rho}\right)^{\frac{1}{\tau}} = O\left(\left(1 - \frac{2}{1+\|\mathbf{A}\|_{2,1}}\right)^{\|\mathbf{A}\|_{2,1}\log(1/\epsilon)}\right) = O(\log(1/\epsilon)). \tag{114}$$

Hence, $O(\log(1/\epsilon))$ bits of precision suffice to implement this method.

# G Proof of Theorem 2

Based on the result of Lemma 6, we can obtain the iteration complexity to solve each subproblem with a given accuracy $\epsilon_k$ and confidence level $p$. To guarantee that $K$ subproblems are all solved to precision $\epsilon_k$ with probability $1 - p$, it suffices each of them to hold with probability $1 - p/K$ (by union bound). Combining the above results and the iteration complexity of ACDM, we have the required number of inner iterations in the each outer iteration is

$$O\left(\sum_{i=1}^{n} \|\mathbf{A}_i\| \log\left(\frac{D_0^k K}{\epsilon_k p}\right)\right) = O\left(\sum_{i=1}^{n} \|\mathbf{A}_i\| \log\left(\frac{\rho(D_0^k)^{\frac{1}{3}}\gamma^2}{\epsilon^{\frac{2}{3}} p^{\frac{1}{3}}} \log\left(\frac{2D_0}{\epsilon}\right)\right)\right). \tag{115}$$

Finally, we estimate the worst-case overall complexity of Algorithm 1. In each iteration of the ACDM, the Algorithm 2 samples each coordinate $i$ with probability distribution

$$p_i = \frac{\sqrt{\|\mathbf{A}_i\|^2 + 1}}{\sum\limits_{j=1}^{n} \sqrt{\|\mathbf{A}_j\|^2 + 1}},$$

and corresponding iteration cost is $O(nnz(\mathbf{A}_i))$ (given in Lemma 6). Thus, the complexity of solving each subproblem 1 is

$$O\left(\sum_{i=1}^{n} \|\mathbf{A}_i\| nnz(\mathbf{A}_i) \cdot \log\left(\frac{\rho(D_0^k)^{\frac{1}{3}}\gamma^2}{\epsilon^{\frac{2}{3}}p^{\frac{1}{3}}} \log\left(\frac{\gamma D_0}{\epsilon}\right)\right)\right). \tag{116}$$

Thus, the worst case complexity is

$$O\left(\gamma^2 \sum_{i=1}^{n} \|\mathbf{A}_i\| nnz(\mathbf{A}_i) \cdot \log(1/\epsilon) \log\left(\frac{\rho(D_0^k)^{\frac{1}{3}}\gamma^2}{\epsilon^{\frac{2}{3}}p^{\frac{1}{3}}} \log\left(\frac{\gamma D_0}{\epsilon}\right)\right)\right)$$

$$\overset{(a)}{=} O\left(\gamma^2 \sum_{i=1}^{n} a_{\max} nnz(\mathbf{A}_i) \cdot \log(1/\epsilon) \log\left(\frac{\rho(D_0^k)^{\frac{1}{3}}\gamma^2}{\epsilon^{\frac{2}{3}}p^{\frac{1}{3}}} \log\left(\frac{\gamma D_0}{\epsilon}\right)\right)\right)$$

$$\overset{(b)}{=} O(a_m \theta_{S^*}^2 (R_x\|\mathbf{A}\| + R_z)^2 nnz(\mathbf{A}) \log^2(1/\epsilon)). \tag{117}$$

The above, (a) utilizes the definition of $a_m = \max_i \|\mathbf{A}_i\|$, (b) is based on the estimation that $\gamma = O(\theta_{S^*}(R_x\|\mathbf{A}\| + R_z))$. Thus, the theorem follows.