[Reviews · NeurIPS 2017]

Reviewer 1



The paper is well written and discusses an interesting problem of linear program method which is applicable to several optimization procedures. It is not completely clear on which data set (experimental setup) FADMM algorithm is evaluated, I would expect at least a small paragraph on this, or providing references to the existing setups.

Reviewer 2



This paper presents a new splitting method approach to the linear programming problem which has theoretical and practical performance advantages over existing approaches. Experiments are given showing the performance gains on a variety of problems that arise from reformulations of machine learning problems. I like the approach taken by this paper. First order approaches to the LP problem are under-studied in my opinion. The most notable papers are from the early 90s. The recent resurgence in the popularity of ADMM style splitting methods in the NIPS community makes this paper relevant and timely. In particular, the method given relies on recent advances in solving well-conditioned unconstrained least-squares problems which were not known the in 90s. The presentation of the paper is generally good. The graphs and tables are legible and explained well. The experiments are more comprehensive than typical NIPS papers. Often papers on the LP problem report results on the old NETLIB suite. They are very different in structure from the ML problems considered in this work, so it's not obvious that the same approaches should work well on both. I think it's fine for the NIPS paper to consider just ML problems; It would be interesting to see results on the NETLIB problems though. There are some minor grammatical issues in the text. I've pointed out a few below. Otherwise well written. “compared with current fastest LP solvers” “linearly at a faster rate compared with the method in [9]” “The work [10] prove that” line 100. “equivalent to solve a” line 123. “Let the gradient (8) vanishes” line 201.

Reviewer 3



This paper develops a novel alternating direction based method for linear programming problems. The paper presents global convergence results, and a linear rate, for their algorithm. As far as I could see, the mathematics appears to be sound, although I did not check thoroughly. Numerical experiments were also presented that support the practical benefits of this new approach; this new algorithm is compared with two other algorithms and the results seem favorable. Note that the authors call their algorithm FADMM - my suggestion is that the authors choose a different acronym because (i) there are several other ADMM variants already called FADMM, and (ii) this is an ADMM for LP so it might be more appropriate to call it something like e.g., LPADMM, which is a more descriptive acronym. The convergence results and numerical experiments are good, but the thing that I liked most about this paper is that the algorithm seems very practical because the first ADMM subproblem can be solved *inexactly*. This is a very nice feature. The authors also put effort to explain how to solve this first subproblem efficiently from a practical perspective (Section 5), which I felt was an important contribution of the paper. One of the reasons that I did not give this paper a higher score is that there are a couple of typos in the paper and the descriptions and language could be clearer in places. For example, $n_b$ is used in (1) but is not defined; $n_f$ is used at the end of Section 2.1 but is not defined; the Hessian in Section 2.1 is not correct; throughout the paper the authors use minus signs instead of hyphens (e.g., $n-$dimensional should be $n$-dimensional, $l_1-$regularized should be $l_1$-regularized etc); the bibliography has several typos especially with $ signs missing: [4] should have $\ell_1$-problems; [11] D and R should be capitalized 'Douglas-Rachford'; [16,17] should be capitalized ADMM. All these are minor, but the bar for NIPS is very high and these should be corrected, and the paper should be thoroughly proofread for English mistakes. Overall I liked this paper.

Reviewer 4



The paper proposes a new splitting method for LP for solving it via ADMM. The algorithm has linear rate of convergence involving only O(m + n) dimensional iterates with convergence constant faster than the current ADMM solver for LP and also weak dependence on problem dimensionality. The paper also explains the slow or fluctuating tail issue that current ADMM based LP solver faces. The inner quadratic problems are well conditioned and can be solved either by direct inversion or using existing accelerated coordinate descent method in case matrix is big but sparse. Results on different datasets show significant speed-ups over vanilla ADMM and current fastest ADMM based LP solver. Overall I liked the paper. The slow tail issue is well observed practical issue with ADMM and theoretical as well as empirical results clearly show the benefits of the proposed approach in solving that. The theoretical proof looks correct and steps are explained well. Experimental section showing results on different ML problems clearly shows the benefit of the approach. One question: typically ADMM approach is very sensitive to augmented lagrangian parameter $\rho$. It would be good to explore the sensitivity of the proposed approach wrt. this parameter.